# Weaving Graph over Tokens: Contextualizing Structured Sequences for LLMs

**Jiaxuan Chen** [* 1 2]  **Zixing Zhang** [* 1 2]  **Ruijun Mao** [1 2]  **Wei Sun** [1 2]  **Zhicheng Liang** [1 2]  **Yuhang Zhang** [1 2]  **Yaxi Liu** [3]  **Fangxin Wang** [1 2 4]

## Abstract

Generative Graph Language Models (GLMs) must reconcile topology with causal language modeling. Linearization obscures multi-hop connectivity, while encoder-based methods bottleneck token-level reasoning during generation. Viewing context modeling as a form of message passing, we introduce **Weaver**, an encoder-free framework that extends the attention mechanism of decoder-only LLMs to enable graph reasoning. Weaver maps graph distances into rotary positional embeddings so that structurally connected nodes become proximate in attention space, propagating information over graph topology as if it were sequential context. To achieve this, we combine: 1) a masking mechanism for causal tokens with graph structures; 2) a unified geometric encoding that couples sequential position and graph distance in joint rotary embeddings (Graph-over-Tokens RoPE); and 3) a design principle to prioritize local information to resolve positional ambiguity under graph symmetries. On zero-shot benchmarks, Weaver achieves state-of-the-art performance among generative GLMs, with gains of up to 30% over prior generative methods on heterophilic graphs, while matching specialized discriminative models on citation networks—all within a unified decoder-only framework.

## 1. Introduction

The integration of Large Language Models (LLMs) with structured data is central to applications ranging from knowledge graph question answering to relational reasoning over attributed networks. While LLMs provide strong semantic priors and flexible generation, extending these capabilities to Text-Attributed Graphs (TAGs)—in which nodes and optionally edges carry rich textual information, interwoven with non-Euclidean relational structure—remains challenging. A primary obstacle is the misalignment between autoregressive sequencing and topology-aware graph aggregation. Closing this gap is necessary for building generative graph language models that can jointly reason about semantics and structure.

A natural baseline is to linearize graphs into node-edge strings and apply an off-the-shelf LLM (Luo et al., 2024; Ye et al., 2024). This approach preserves the LLM interface and benefits from existing pretraining, and can capture direct edge connectivity explicitly listed in the text. However, it weakens structural inductive bias for complex topological patterns: multi-hop relationships, cycles, and graph-level symmetries must be reconstructed from scattered token spans rather than being directly encoded in the attention pattern. As a result, linearization often struggles with tasks requiring reasoning over indirect connectivity or sensitivity to global structural properties.

To incorporate structural inductive bias, many recent GLMs introduce auxiliary graph encoders—typically graph neural networks (GNNs) or graph transformers (GTs)—that compress topological structure together with rich semantic information into static embeddings to condition the LLM's generation (Chai et al., 2023; Tang et al., 2024; Plenz & Frank, 2024; Wang et al., 2024; Chen et al., 2024; Kong et al., 2025). While effective at capturing topology, encoder-based designs create a separation between structural processing and generation: fine-grained graph information is compressed into static embeddings before the LLM performs token-level reasoning. This induces an information bottleneck and a representation mismatch between geometric encodings and the LLM's semantic space, often requiring specialized alignment layers and limiting the LLM's ability to use structure dynamically during generation.

---

[*]Equal contribution [1]School of Science and Engineering (SSE), The Chinese University of Hong Kong (Shenzhen), Shenzhen, China [2]Shenzhen Future Network of Intelligence Institute (FNii-Shenzhen), Shenzhen, China [3]School of Computer and Communication Engineering, University of Science and Technology Beijing, Beijing, China [4]Guangdong Provincial Key Laboratory of Future Networks of Intelligence, The Chinese University of Hong Kong (Shenzhen), Shenzhen, China. Correspondence to: Fangxin Wang <wangfangxin@cuhk.edu.cn>.

*Proceedings of the 43rd International Conference on Machine Learning*, Seoul, South Korea. PMLR 306, 2026. Copyright 2026 by the author(s).

We posit that external encoders are not required for expressiveness: they impose an information bottleneck by compressing structural signals into a fixed-size representation before the LLM consumes them (Alon & Yahav, 2021). Conceptually, a Transformer layer performs attention-based message passing on a fully connected graph (Joshi, 2025), meaning decoder-only LLMs already implement a generic graph computation primitive. The effective limitation is not model capacity, but the *1D sequential bias* imposed by standard causal masking and linear positional encodings, which force topology into a fixed order. By reparameterizing attention visibility and positional structure to align with the input graph, an LLM can be adapted for joint semantic-structural reasoning without auxiliary encoders.

Based on this insight, we propose Weaver, an encoder-free framework that integrates graph topology directly into the LLM attention mechanism. Instead of linearizing structure or precomputing graph embeddings, Weaver represents graph textual elements (nodes and, optionally, edges) as token parallel blocks and modifies attention so that token-level computation performs both *intra-node* semantic encoding and *inter-node* topology-aware information propagation.

Weaver achieves *robust graph modeling* while preserving the *sequential modeling* capabilities of pre-trained LLMs (Qwen3-8B in the paper). Concretely, the framework introduces three streamlined designs: (i) **Graph-Causal Masking** enables global visibility across graph elements while enforcing causality within each node's local text sequence; (ii) **Graph-over-Tokens RoPE (GoRoPE)** unifies positional encoding by modulating attention based on both relative token positions and relative graph distances, allowing the model to jointly respect semantic order and structural topology; and (iii) **Local Prioritization Principle** resolves the language ambiguity under graph symmetries by ensuring that topologically equivalent nodes remain distinguishable during attention computation.

We empirically validate our new framework on a suite of synthetic and graph-language modeling benchmarks.

- **More Expressive Inductive Bias.** Weaver injects topological priors into attention, enabling multi-hop structural comprehension. On synthetic graph regression tasks (e.g., Weighted Shortest Path, Monochromatic Subgraph), Weaver avoids the performance collapse of Qwen3–8B and the state-of-the-art GLM baseline (GOFA) at 3-hop depth, and the advantage grows as graph size scales up.
- **Stronger Generalization.** Weaver significantly outperforms generative GLM baselines in zero-shot transfer on both homophilic and heterophilic graphs, suggesting that topology encoded via relative positional encoding generalizes more robustly than learned message passing.
- **New Flexibilities.** Weaver supports free-form graph QA by dynamically attending to relevant structures defined

in natural language prompts, removing the need for pre-specified target nodes. Additionally, Weaver's positional encoding scheme naturally handles both structural-only and text-attributed edges within a unified framework.

## 2. Preliminaries

**Graph Laplacian and Positional Encodings (PEs).** For graph $\mathcal{G}$ with adjacency matrix $\boldsymbol{A}$ and degree matrix $\boldsymbol{D}$, the normalized Laplacian is:

$$\boldsymbol{L}_{\text{sym}} = \boldsymbol{I} - \boldsymbol{D}^{-1/2}\boldsymbol{A}\boldsymbol{D}^{-1/2} = \boldsymbol{U}\boldsymbol{\Lambda}\boldsymbol{U}^{\top}, \quad (1)$$

where $\boldsymbol{U}$ contains eigenvectors and $\boldsymbol{\Lambda}$ eigenvalues. Laplacian Position Encodings (LapPE) (Defferrard et al., 2016; Dwivedi & Bresson, 2021; Dwivedi et al., 2023) assign node $u$ the $u$-th row of $\boldsymbol{U}$ restricted to the $p$ smallest non-trivial eigenvalues, yielding $\boldsymbol{\phi}_u \in \mathbb{R}^p$. Such spectral features encode structural proximity and are widely used in node and edge embeddings. While these capture rich topological information, eigenvectors are only defined up to sign and, for repeated eigenvalues, up to basis rotation within the eigenspace.

**Attention Masking.** In decoder-only LLMs, multi-head attention applies a causal mask to enforce autoregressive token generation. For tokens at positions $m$ and $n$, the standard causal mask is defined as

$$M(m, n) = \begin{cases} 0 & \text{if } n \leq m, \\ -\infty & \text{otherwise,} \end{cases} \quad (2)$$

which is added to attention logits before softmax, ensuring each token attends only to itself and preceding tokens.

**Rotary Position Embedding (RoPE).** RoPE (Su et al., 2021) encodes relative positions via rotations in orthogonal 2D subspaces. For token at position $m$ with embedding $\boldsymbol{x}_m \in \mathbb{R}^d$, RoPE transforms queries and keys as:

$$f_{\boldsymbol{W},\Theta}(\boldsymbol{x}_m, m) := \bigoplus_{k=1}^{d/2} \rho(m\theta_k)[\boldsymbol{W}\boldsymbol{x}_m]_{2k-2:2k-1}, \quad (3)$$

where $\boldsymbol{W} \in \{\boldsymbol{W}_q, \boldsymbol{W}_k\}$ are projection matrices, $\bigoplus$ denotes direct sum (block concatenation), $\Theta = \{\theta_k\}_{k=1}^{d/2}$ are frequency parameters, and

$$\rho(\theta) = \begin{pmatrix} \cos\theta & -\sin\theta \\ \sin\theta & \cos\theta \end{pmatrix} \quad (4)$$

is the 2D rotation matrix. The key property of RoPE is that the inner product between queries and keys depends only on relative position:

$$\langle f_{\boldsymbol{W}_q,\Theta}(\boldsymbol{x}_m, m), f_{\boldsymbol{W}_k,\Theta}(\boldsymbol{x}_n, n) \rangle = g(\boldsymbol{x}_m, \boldsymbol{x}_n, m - n), \quad (5)$$

enabling efficient relative positional encoding LLMs.

# 3. Methodology

We introduce Weaver (Figure 1), which adapts a pretrained causal LLM to TAGs by aligning language attention to TAGs in three complementary ways. First, *Graph-Causal Masking* changes *visibility* by permitting bidirectional attention across graph elements while preserving autoregressive causality within each element. Second, *GoRoPE* changes *geometry* by injecting graph positional information into the rotary phase. Third, *Local Prioritization* adds a soft inductive bias that favors local context when structural cues are ambiguous. Together, these modifications enable graph-conditioned generation without an encoder.

## 3.1. Input Representation and Token Attributes

**Text-Attributed Graphs (TAGs).** We define a TAG as $\mathcal{G} = (\mathcal{V}, \mathcal{E}, \{\mathbf{t}_u\}_{u \in \mathcal{U}})$, where $\mathcal{U} := \mathcal{V} \cup \mathcal{E}$ is the set of structural elements, and each element $u$ (node or edge) carries a token sequence $\mathbf{t}_u = (t_{u,1}, \ldots, t_{u,L_u})$.

**Unified Formulation.** We adopt a single graph-conditioned autoregressive formulation for node-, link-, and graph-level QA. Given a TAG $\mathcal{G}$, a user query $\mathbf{q} = (q_1, \ldots, q_M)$, and a target response $\mathbf{y} = (y_1, \ldots, y_T)$, the model defines

$$p(\mathbf{y} \mid \mathcal{G}, \mathbf{q}) = \prod_{t=1}^{T} p(y_t \mid \mathbf{y}_{<t}, \mathcal{G}, \mathbf{q}). \quad (6)$$

These tasks differ only in which part of $\mathcal{G}$ the query refers to: a node, a node pair, or the whole graph.

**Input Stream.** To process $\mathcal{G}$ in a single Transformer pass, we serialize the text-attributed graph (nodes and edges) and the conversation history (system prompts, user queries) into a single packed sequence of tokens. Figure 2 shows the chat template used for serialization, where edge descriptions are only included beyond connectivity.

**Token Attributes.** To preserve structure while flattening the input, we assign each token $i$ in the sequence a tuple of positional attributes $(m_i, \mathbf{\Psi}_i)$:

- **Local Position ($m_i$).** The index of token $i$ within its specific source element (e.g., the $m$-th word of a node's text). This counter resets to 0 for each new node, edge, or conversation turn[1].
- **Structural PE ($\mathbf{\Psi}_i$).** A geometric encoding $\mathbf{\Psi}_i \in \mathbb{R}^p$ derived from the node-level PE (defined in §3.2.1).

**Logical Indexing.** For the derivation of the attention mechanism, it is convenient to reference tokens by their logical origin rather than their linear position in the packed sequence. Hereafter, let $b$ represent a generic input element (a node,

---

[1]In practice, graph inputs may include a shared prefix (e.g., system prompt or task description) before the structured elements. We omit this prefix for notational simplicity.

an edge, or a conversation turn). We denote the $m$-th token of element $b$ as $t_{b,m}$. Because the masking and GoRoPE operations depend solely on these logical indices $(b, m)$—and not on the absolute position in the packed sequence—the model is invariant to the reordering of blocks.

## 3.2. Weaver: Graph-Language Attention

Pretrained LLMs use RoPE to model relative distance in text sequences: tokens far apart receive decayed attention. We extend this principle to graphs: tokens in structurally distant elements should receive decayed attention based on graph topology. Weaver modifies standard multihead attention in two components: **Graph-Causal Masking** determines *visibility*, and **GoRoPE** encodes *relative topological distance*.

**Graph-Causal Masking.** Weaver employs a hybrid mask that reconciles sequential text dependencies with graph connectivity (Figure 1, bottom-left). *Intra-block* attention preserves autoregressive causality: tokens within the same block attend only to predecessors. *Inter-block* attention is bidirectional for graph elements, enabling global message passing analogous to Graph Transformers (Dwivedi & Bresson, 2021; Ma et al., 2023b).

Formally, let $b_i$ denote the block identifier of token $i$. For graph tokens, the attention mask $M_{i,j} \in \{0, -\infty\}$ is:

$$M_{i,j} = \begin{cases} 0 & \text{if } b_i = b_j \text{ and } m_j \leq m_i \text{ (Causal)}, \\ 0 & \text{if } b_i \neq b_j \text{ and } b_i, b_j \in \mathcal{G} \text{ (Global)}, \\ -\infty & \text{otherwise}. \end{cases} \quad (7)$$

Tokens belonging to the conversation history (user queries/responses) apply standard causal masking to all preceding tokens in the packed sequence.

**Graph-over-Tokens RoPE (GoRoPE).** To encode topology within the attention mechanism, we extend RoPE with structural coordinates $\mathbf{\Psi}$. Crucially, GoRoPE is a **per-token** operation. For token $i$ at frequency band $k$, we define the absolute phase shift $\Theta_{i,k}$:

$$\Theta_{i,k} = m_i \theta_k + s \cdot \Psi_{i,\ell} \theta_k, \quad (8)$$

where $\ell = k \bmod p$ indexes the spectral coordinate and $s$ scales the structural influence. The embedding function rotates the query/key vector $\boldsymbol{x}_i$ by this absolute phase:

$$f_{\boldsymbol{W}}(\boldsymbol{x}_i, m_i, \mathbf{\Psi}_i)_k = \mathcal{R}(\Theta_{i,k})[\boldsymbol{W}\boldsymbol{x}_i]_{2k-2:2k-1}. \quad (9)$$

Since $\mathcal{R}(\alpha)^\top \mathcal{R}(\beta) = \mathcal{R}(\alpha - \beta)$, the inner product $\mathbf{q}_i^\top \mathbf{k}_j$ is driven by the relative difference $\Delta\Theta_{i,j,k} = (m_i - m_j)\theta_k + s(\Psi_{i,\ell} - \Psi_{j,\ell})\theta_k$, recovering the pairwise structural distance without computational overhead.

**Context Modulation.** By interpreting the phase difference, GoRoPE maps graph distances into the LLM's sequential

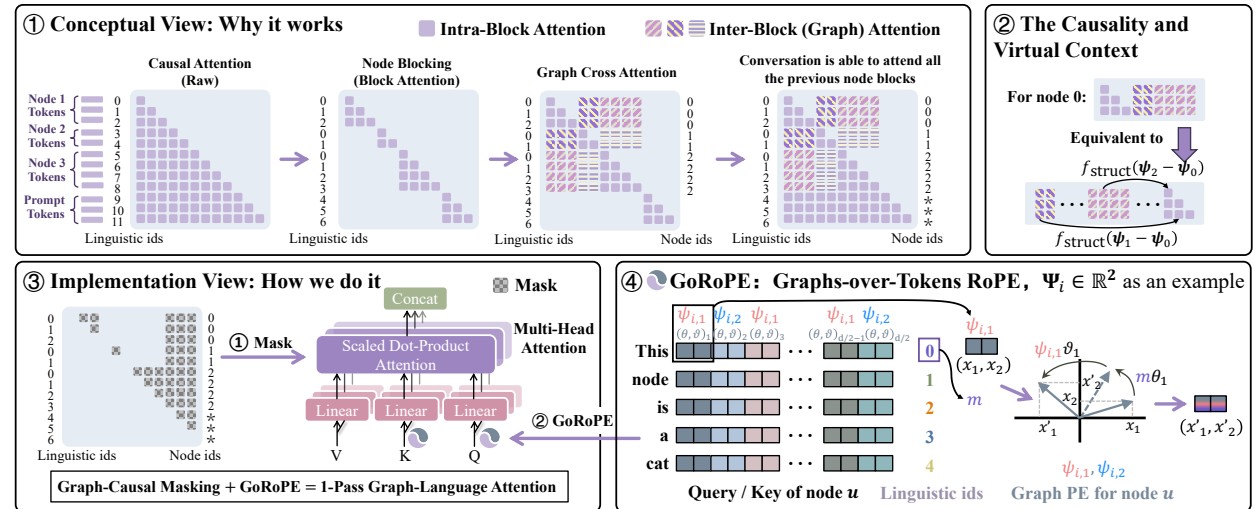

*Figure 1.* **Overview of Weaver.** Weaver injects text-attributed graph topology directly into decoder-only LLM attention without external graph encoders. The graph and conversation are packed into one sequence, while each token retains a block-local position and a structural coordinate. **(1)** Weaver replaces arbitrary global token serialization with graph-element blocks, shown here as node blocks. Attention is causal within each block and bidirectional across graph blocks, while conversation tokens follow standard causal masking and attend to preceding graph context. **(2)** Cross-block attention can be viewed as attending to a virtual graph context defined by relative structural offsets, rather than by the order in which blocks are serialized. **(3)** This attention pattern is implemented in a single Transformer pass using Graph-Causal Masking for visibility control and GoRoPE for geometry-aware query/key rotation. **(4)** GoRoPE extends RoPE by coupling block-local token positions with graph positional coordinates, mapping structural proximity into the pretrained rotary attention space for topology-aware generation.

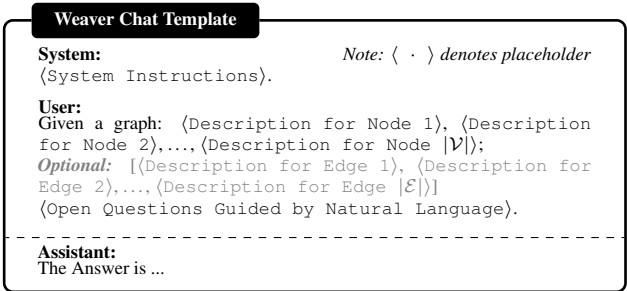

*Figure 2.* Weaver chat template. Edge descriptions are optional and included only when they possess semantic information beyond connectivity (e.g., relationship types or weights). Explicit connectivity details are omitted because Weaver directly encodes the graph structure with GoRoPE.

attention space. The effective attention score becomes:

$$A_{i,j} = (\boldsymbol{W}_Q \boldsymbol{x}_i)^\top \mathcal{R}\left(\theta_k[\Delta m_{ij} + s \cdot \Delta \Psi_{ij,\ell}]\right)(\boldsymbol{W}_K \boldsymbol{x}_j). \quad (10)$$

Structurally distant nodes induce large phase shifts, effectively separating them in the rotary space. This allows the model to apply its pretrained long-context decay bias to graph topology without retraining.

### 3.2.1. UNIFIED ENCODING CONSTRUCTION

We construct the structural coordinates $\boldsymbol{\Psi}_i$ to unify nodes, edges, and conversation text into a common geometric space. We first compute Laplacian positional encodings

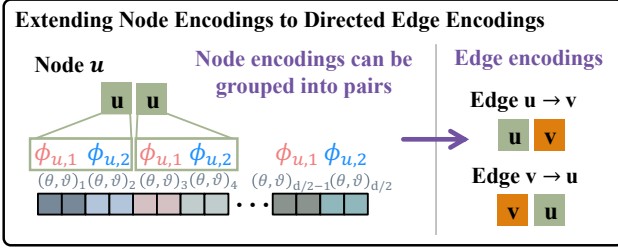

*Figure 3.* Encoding directed-edge attributes from node encodings. Node encodings are grouped in pairs and each edge $u \to v$ composes encodings from both endpoints, enabling directional topology within our positional encoding scheme.

(LapPE) (Dwivedi et al., 2023) $\boldsymbol{\phi}_u \in \mathbb{R}^{p/2}$ for every node $u \in \mathcal{V}$. Following standard practice in spectral graph theory, Weaver adopts canonized LapPE (Ma et al., 2023a) for robustness to Laplacian sign ambiguity. We then assign $\boldsymbol{\Psi}_i \in \mathbb{R}^p$ based on the source element $b_i$ of token $i$:

$$\boldsymbol{\Psi}_i = \begin{cases} [\boldsymbol{\phi}_u^\top, \boldsymbol{\phi}_u^\top]^\top & \text{if } b_i \text{ is node } u, \\ [\boldsymbol{\phi}_u^\top, \boldsymbol{\phi}_v^\top]^\top & \text{if } b_i \text{ is edge } u \to v, \\ [\bar{\boldsymbol{\phi}}^\top, \bar{\boldsymbol{\phi}}^\top]^\top & \text{if } b_i \text{ is conversation text,} \end{cases} \quad (11)$$

where $\bar{\boldsymbol{\phi}}$ is the centroid of node embeddings, ensuring instruction tokens remain structurally *unbiased*. For edges $(u \to v)$, we construct $\boldsymbol{\Psi}_i$ by concatenating the source and target node embeddings (Figure 3). By allocating $\boldsymbol{\phi}_u$ and $\boldsymbol{\phi}_v$ to disjoint frequency bands (via the cyclic index $\ell$), GoRoPE

allows the attention mechanism to distinguish edge directionality ($\boldsymbol{\Psi}_{u\to v} \neq \boldsymbol{\Psi}_{v\to u}$) without requiring separate edge identifiers. This design aligns with TokenGT (Kim et al., 2022), which demonstrates that concatenating endpoint identifiers $[\boldsymbol{P}_u, \boldsymbol{P}_v]$ enables expressive graph reasoning in Transformers. It is also aligned with multidimensional rotary encodings such as MRoPE (Qwen Team, 2024), which assign different spatial axes to different channel groups; here, "source" and "target" act as two topological **bases** that the attention mechanism can empirically distinguish.

### 3.2.2. SPECTRAL COHERENCE AND STRUCTURAL SCALE $s$

The scalar $s$ controls how structural offsets map to the rotational space. Since $\theta_k$ decays exponentially across bands, large structural distances $|\Delta\Psi_{ij}|$ induce rapid oscillations (loss of coherence) in high-frequency heads first, effectively gating attention to local neighborhoods. Low-frequency heads remain coherent over longer structural distances. To ensure the lowest-frequency bands—which represent the longest context—remain coherent and free of phase wraparound (aliasing), we bound the maximum structural offset to the backbone's effective context limit $L_{\max}$:

$$s \approx \frac{L_{\max}}{\|\Delta\Psi\|_{\max}}, \quad \|\Delta\Psi\|_{\max} := \max_{i,j}\|\boldsymbol{\Psi}_i - \boldsymbol{\Psi}_j\|_{\infty}. \tag{12}$$

This allows the model to interpret structural distances as valid long-range dependencies within the pretrained attention capacity. See Appendix B for a detailed discussion.

### 3.2.3. LOCAL PRIORITIZATION

GoRoPE is defined through relative structural phases. Hence, blocks that are indistinguishable under the structural positional encoding can yield $\Delta\Psi_{ij} = \boldsymbol{0}$, making the corresponding cross-block tokens positionally indistinguishable from tokens in the query block. We therefore add a lightweight block-level bias to the attention logits:

$$\Delta_{i,j,h} = \begin{cases} -\gamma\beta_h & \text{if } b_i \neq b_j \text{ and } \{i,j\} \cap \mathcal{C} = \emptyset, \\ 0 & \text{otherwise,} \end{cases} \tag{13}$$

where $b_i$ denotes the block containing token $i$, $\mathcal{C}$ denotes global tokens such as system prompts, $\gamma \geq 0$ controls the bias strength, and $\beta_h$ is a head-dependent coefficient chosen according to the ALiBi per-head schedule (Press et al., 2022). The bias therefore does not penalize two connected nodes more strongly because they appear far apart in the serialized sequence. Instead, it softly prioritizes intra-block token interactions while keeping cross-block attention available whenever content-based or structure-based attention scores are sufficiently strong. Design alternatives are discussed in Appendix B.3, and ablations are reported in §5.6.

### 3.3. Graph-Language Alignment and Task Adaptation

**Aligning GoRoPE and Directed Edge Encoding.** To inject fundamental graph cognition, we introduce a structural alignment stage using two data sources: (i) 300k synthetic graphs (max 50 nodes, edge prob. 0.3) covering core reasoning primitives—Shortest Path, Weighted SP, Common Neighbors, and Monochromatic Subgraph (Appendix C.2); (ii) 50k SceneGraph samples to reinforce directed edge understanding (Appendix C.1). Treating GoRoPE as a positional encoding extension, we apply LoRA for 1 epoch (Appendix C.5). The resulting **Weaver-Zero** exhibits emergent directed edge recognition on unseen domains and achieves competitive zero-shot performance (Table 2) with 62 GPU hours.

**Task-Specific Finetuning.** To ensure a fair comparison with state-of-the-art baselines (e.g., GOFA), we subsequently fine-tune Weaver on standard node classification datasets (Arxiv and Pubmed, 28k graphs). We train for 1 epoch (68 GPU hours), strictly aligning our supervision protocol with established baselines.

## 4. Related Work

**Graph Linearization.** Approaches that linearize graph structures into text prompts leverage the native reasoning of off-the-shelf LLMs (Zhao et al., 2023; Luo et al., 2024; Ye et al., 2024). However, they often fail to capture complex topological dependencies.

**Encoder-Enhanced Graph LLMs.** To better capture structural information, many recent works introduce auxiliary graph encoders (e.g., GNNs) that align graph embeddings with the LLM's token space (Liu et al., 2024; Chen et al., 2024; Tang et al., 2024; Kong et al., 2025; He et al., 2025). For instance, GraphGPT (Tang et al., 2024) and Graph-Translator (Zhang et al., 2024) employ a projector to map outputs from a pre-trained GNN into the LLM's input space. LLaGA (Chen et al., 2024) utilizes a projector to map neighborhood invariants into soft prompts. GOFA (Kong et al., 2025) and UniGraph (He et al., 2025) further explore co-training strategies for these encoders. While effective, these architectures create an information bottleneck: fine-grained structural nuances are compressed into static embeddings before reaching the LLM, decoupling the message-passing process from token-level generation.

**Graph Positional Encodings.** Graph Transformers inject structural bias via positional encodings (Buterez et al., 2025; Dwivedi & Bresson, 2021; Rampášek et al., 2022), ranging from spectral features to random-walk distances (Ma et al., 2023b). Recent work extends rotary PE to graphs, with WIRE (Reid et al., 2026) using learnable spectral relaxations of RoPE (Su et al., 2021). However, these approaches typically assume fully observed graphs and train GNNs/Graph

Table 1. TAG Datasets selected in experiments.

| Dataset | Task | Structure |
|---|---|---|
| **Homophilic Graphs** | | |
| Cora (McCallum et al., 2000) | N&L | Hom. |
| Pubmed (Sen et al., 2008) | N&L | Hom. |
| Arxiv (Hu et al., 2020) | N&L | Hom. |
| Products (Hu et al., 2020) | N&L | Hom. |
| Citeseer (Giles et al., 1998) | NC | Hom. |
| History (Ni et al., 2019) | NC | Hom. |
| Children (Ni et al., 2019) | NC | Hom. |
| Sportsfit (Ni et al., 2019) | NC | Hom. |
| WikiCS (Mernyei & Cangea, 2020) | NC | Hom. |
| ExplaGraphs (Saha et al., 2021) | GQA | Hom. |
| SceneGraphs (Hudson & Manning, 2019) | GQA | Hom. |
| **Heterophilic Graphs** | | |
| Cornell (Craven et al., 1998) | NC | Het. |
| Texas (Craven et al., 1998) | NC | Het. |
| Wisconsin (Craven et al., 1998) | NC | Het. |
| Washington (Craven et al., 1998) | NC | Het. |

*Note:* N&L = Node Classification & Link Prediction, NC = Node Classification, GQA = Graph Question Answering, Hom. = Homophilic, Het. = Heterophilic.

Transformers from scratch for a fixed set of tasks, which limits transfer across domains and prevents leveraging the semantic knowledge and instruction-following abilities of pre-trained LLMs. In contrast, Weaver aligns rotary encodings with the LLM language space to inject structural priors into pre-trained language models, enabling more general graph reasoning without task-specific retraining.

Weaver reconciles encoder-based methods (structural bias but limited granularity) with linearization (full tokens but no topology) via attention-only modifications. See Appendix A for architectural comparisons.

# 5. Empirical Evaluation

We evaluate Weaver on graph reasoning and regression tasks to address the following questions:

- **RQ1:** Can Weaver generalize to unseen graph tasks without task-specific training?
- **RQ2:** Does Weaver improve over baselines when fine-tuned on graph-specific tasks?
- **RQ3:** Does Weaver genuinely encode and utilize graph structure during generation?
- **RQ4:** What are the computational costs of Weaver in terms of time and memory?

## 5.1. Setup

As shown in Table 1, we evaluate Weaver on 15 text-attributed graph benchmarks spanning node classification (NC), link prediction (LP), and graph question answering (Graph QA). We first adapt the backbone to graph-structured inputs using four synthetic structural reasoning tasks (§3.3) yielding Weaver-Zero. For downstream adaptation, following GOFA (Kong et al., 2025), we train on Pubmed and ogbn-Arxiv with supervised samples of 40000/80000 in-

stances for Arxiv (LP)/Arxiv (NC) and 10000/10000 instances for Pubmed (LP)/Pubmed (NC), obtaining Weaver. We use the Wang et al. (2025a) protocol for data preprocessing, prompting, and evaluation, and ensure that all baselines are compared under the same setting. All experiments use Qwen3-8B as the backbone and are conducted on $8 \times$ A100-80GB GPUs. We compare against 11 task-specific methods and 5 generative approaches; detailed dataset statistics, preprocessing, and baseline descriptions are provided in Appendix C.

## 5.2. Zero-shot Transfer Across Graph Tasks

To answer **RQ1**, we evaluate zero-shot node classification on both homophilic and heterophilic graphs (Table 2). For Weaver-Zero, all test datasets are unseen; for Weaver, all test datasets except Pubmed and ogbn-Arxiv are unseen. Additional analysis on the effect of label set size is provided in Appendix D.1.

**Versus Specialized Methods.** Among *specialized* baselines, LLM-BP performs best overall by combining LLM embeddings with belief propagation and an explicit homophily estimator, yielding strong results on heterophilic graphs (e.g., 83.3% on Cornell). However, these specialized pipelines are designed specifically for fixed tasks and do not naturally extend to other graph-language tasks (e.g., open-ended QA or structured description). In contrast, Weaver can be applied under a unified prompting interface across tasks. Despite this broader applicability, Weaver matches or exceeds LLM-BP on homophilic benchmarks (e.g., 76.5% vs. 72.6% on Cora). On heterophilic graphs, Weaver is 10–20 points below LLM-BP; one possible explanation is that LLM-BP explicitly models homophily, whereas GoRoPE primarily encodes spectral proximity. Nevertheless, Weaver outperforms the other task-specialized baselines on the four heterophilic datasets, achieving a macro-average of 61.3% over Cornell/Texas/Wisconsin/Washington.

**Versus Generative Methods.** Encoder-based GLMs (GraphGPT, LLaGA, GOFA) pass fixed encoder outputs to the decoder, exposing structural information only through this bottleneck. Empirically, GOFA attains 71.1% on Cora but averages 35.4% across the four heterophilic datasets. Under the same evaluation, Weaver, which incorporates topology into attention via GoRoPE, exceeds GOFA by an average of +10.5 points on homophilic datasets (range: +1.2 to +25.7; excluding Pubmed) and +26.0 points on heterophilic datasets (range: +21.5 to +30.0). To further isolate the contribution of the underlying language model, we include *Qwen3-8B-FT*, a fine-tune of Weaver's base LM under Wang et al. (2025a)'s prompting setup for textual methods. Weaver outperforms QWEN3-8B-FT on most benchmarks, with the largest gap on Cornell (+21 points), suggesting that the observed gains are not attributable to the base LM alone.

*Table 2.* Zero-shot node classification accuracy (%, ↑) on homophilic and heterophilic graphs. **Bold**: best in group; underline: second-best. *Weaver* is a generative method; we include it in both groups for comprehensive comparison. Since Pubmed is included in Weaver's training data, we exclude it from Weaver's evaluation for fair comparison (Weaver's average is computed over 10 datasets). Avg. Rank is computed within each method group (Specialized or Generative). Weaver achieves the strongest average rank among generative methods and remains competitive with specialized methods.

| | HOMOPHILIC | | | | | | | HETEROPHILIC | | | | |
| METHOD | Cora | Citeseer | Pubmed | History | Children | Sportsfit | WikiCS | Cornell | Texas | Wisconsin | Washington | Avg. Rank |
|---|---|---|---|---|---|---|---|---|---|---|---|---|
| ***Specialized Methods*** | | | | | | | | | | | | |
| SBERT | 69.75 | 66.69 | 70.57 | 53.53 | 22.59 | 43.79 | 59.06 | 63.66 | 64.58 | 62.10 | 63.52 | 6.45 |
| SBERT + NA | 72.49 | 68.66 | 71.26 | 57.86 | 25.28 | 46.84 | 66.26 | 54.21 | 56.04 | 54.23 | 58.88 | 5.45 |
| DGI | 16.79 | 15.24 | 25.10 | 20.98 | 2.22 | 7.48 | 14.98 | 14.66 | 11.23 | 12.08 | 20.96 | 12.45 |
| RoBERTa | 70.71 | 66.95 | 69.54 | 55.39 | 24.25 | 41.51 | 59.08 | 61.68 | 62.25 | 60.33 | 60.60 | 6.45 |
| GraphMAE | 15.13 | 8.11 | 36.56 | 36.36 | 7.24 | 30.50 | 8.91 | 23.04 | 17.65 | 23.02 | 24.89 | 11.00 |
| UniGLM | 45.57 | 52.26 | 70.33 | 44.24 | 21.48 | 33.46 | 55.05 | 23.03 | 21.39 | 27.16 | 24.01 | 10.00 |
| ZeroG | 60.40 | 50.35 | 74.68 | 36.55 | 12.72 | 14.27 | 46.74 | 10.47 | 53.48 | 12.66 | 8.30 | 10.27 |
| OFA | 20.36 | 41.31 | 28.18 | 8.25 | 3.05 | 15.18 | 30.77 | 29.84 | 11.77 | 4.80 | 6.04 | 12.55 |
| Text-Embed-3-Large | 71.90 | 66.24 | 75.96 | 50.15 | 24.68 | 58.39 | 61.78 | 81.50 | 75.42 | 73.14 | 66.35 | 4.73 |
| LLM2Vec | 67.34 | 67.13 | 74.57 | 53.14 | 25.56 | 57.00 | 62.34 | 81.26 | 76.68 | 73.36 | 65.92 | 4.55 |
| LLM-BP | 72.59 | 69.51 | 75.55 | 59.86 | 24.81 | 61.92 | 67.75 | **83.28** | **81.66** | **77.75** | **73.14** | **2.00** |
| **Weaver-Zero** | 58.20 | 40.00 | **76.80** | 54.40 | 26.60 | 51.20 | 66.80 | 52.30 | 44.90 | 52.20 | 46.00 | 6.00 |
| **Weaver** | **76.50** | **70.14** | – | **60.60** | **34.30** | **63.60** | **69.80** | 69.40 | 60.90 | 62.50 | 52.50 | 2.90 |
| ***Generative Methods*** | | | | | | | | | | | | |
| GPT-3.5-Turbo | 70.11 | 66.83 | 89.75 | 55.07 | 29.73 | **67.21** | 65.53 | 45.54 | 56.14 | 58.86 | 51.09 | 3.27 |
| GPT-4o | 70.29 | 64.77 | **89.85** | 53.30 | 30.76 | 66.35 | 66.10 | 45.54 | **63.10** | 56.60 | 48.90 | 3.18 |
| Qwen3-8B-FT | 70.70 | 63.67 | – | 52.33 | 30.67 | 60.67 | 66.00 | 48.16 | 62.03 | 60.75 | 51.09 | 3.50 |
| GraphGPT | 17.48 | 13.93 | 42.94 | 12.31 | 9.94 | 4.53 | 33.59 | 10.18 | 18.48 | 12.35 | 20.64 | 7.18 |
| LLAGA | 11.62 | 19.52 | 7.56 | 7.95 | 10.09 | 1.84 | 10.98 | 12.57 | 15.51 | 15.09 | 10.48 | 7.45 |
| GOFA | 71.06 | 65.72 | 74.76 | 56.25 | 12.15 | 37.87 | 68.62 | 39.50 | 38.37 | 32.51 | 31.02 | 4.45 |
| **Weaver-Zero** | 58.20 | 40.00 | 76.80 | 54.40 | 26.60 | 51.20 | 66.80 | 52.30 | 44.90 | 52.20 | 46.00 | 4.45 |
| **Weaver** | **76.50** | **70.14** | – | **60.60** | **34.30** | 63.60 | **69.80** | **69.40** | 60.90 | **62.50** | **52.50** | **1.40** |

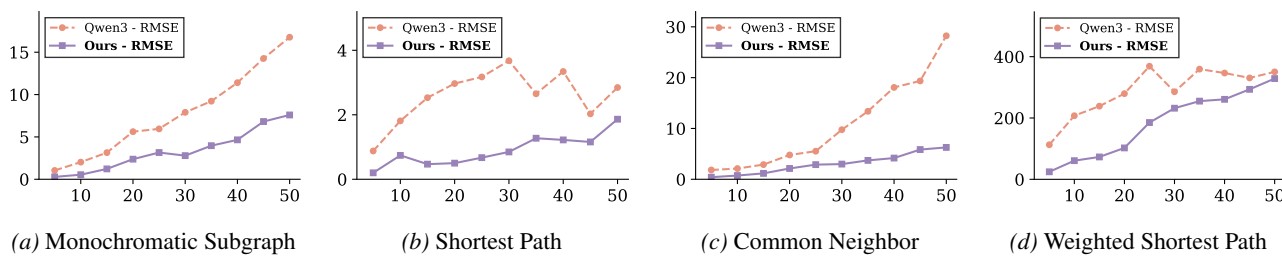

*(a)* Monochromatic Subgraph  *(b)* Shortest Path  *(c)* Common Neighbor  *(d)* Weighted Shortest Path

*Figure 4.* RMSE (↓) on four algorithmic reasoning tasks as graph size (number of nodes) increases. Ours consistently outperforms Qwen3 across all tasks, with the performance gap widening on larger graphs.

*Table 3.* *Weaver* results on the test splits of training datasets. (Accuracy in %, ↑).

| Method | Pubmed (NC) | Arxiv (NC) | Pubmed (LP) | Arxiv (LP) |
|---|---|---|---|---|
| GOFA | 83.83 | 74.77 | 93.97 | 85.41 |
| LLaGA | 81.56 | 74.46 | 89.81 | **92.04** |
| **Weaver** | **86.20** | **85.90** | **94.20** | 88.10 |

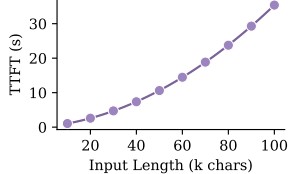

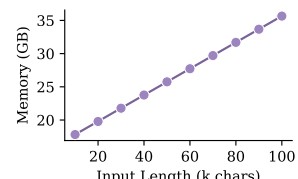

*(a)* Latency (Time to First Token, TTFT) scales quadratically with input length.  *(b)* Memory usage scales linearly with input length.

*Figure 5.* Scaling behavior of our method.

A similar trend holds against GPT-4o, consistent with the benefit of an explicit structural inductive bias.

### 5.3. Results on Adapted Tasks

To answer **RQ2**, we train Weaver on Pubmed and ogbn-Arxiv for node classification (NC) and link prediction (LP). As shown in Table 3, Weaver achieves the best performance in three of four settings: 86.2% on Pubmed NC (+2.4 over GOFA), 85.9% on Arxiv NC (+11.1 over GOFA), and 94.2% on Pubmed LP. On Arxiv LP, LLaGA (92.0%) outperforms Weaver (88.1%), which is consistent with LLaGA being

optimized for pairwise link representations.

These results suggest that Weaver's improvements in zero-shot transfer do not come at the expense of supervised adaptation. Across both datasets, Weaver benefits from using GoRoPE as a structural prior during fine-tuning, enabling consistent gains on node classification and competitive performance on link prediction. The larger improvement on Arxiv NC further indicates that injecting topology into attention remains effective when adapting to larger graphs, where encoder-compressed representations may be less flexible.

## 5.4. Structural Reasoning and SceneGraph QA

To answer **RQ3**, we move beyond fixed-label classification to two generation tasks that jointly stress structural reasoning and language production. The first is synthetic structural reasoning, whose targets are determined purely by graph topology and thus isolate the model's ability to reason over structure. The second is SceneGraph question answering on GQA (Hudson & Manning, 2019), whose answers hinge on the connectivity among visual entities and therefore require combining graph-level reasoning with free-form language generation. Together, these tasks examine whether Weaver augments its base LLM with structural competence without compromising linguistic capability. We report RMSE and accuracy for the quantitative study below; see Appendix D.3 for qualitative case studies of their generation results.

**Structural Reasoning Across Graph Scales.** We compare Weaver with its base LLM (Qwen3) on synthetic tasks whose targets are fully determined by graph topology, and report RMSE as graph size grows from 5 to 50 nodes (Figure 4; full results in Appendix C.2). Across shortest path (SP), weighted shortest path (WSP), common neighbor (CN), and monochromatic subgraph (MS), Weaver consistently attains lower RMSE, and the gap generally widens with scale. At 50 nodes, Weaver reduces RMSE from 2.85 to 1.86 on SP, from 350.43 to 328.55 on WSP, from 28.24 to 6.27 on CN ($4.5\times$), and from 16.77 to 7.59 on MS ($2.2\times$). The largest gains arise on CN and MS, which require multi-hop aggregation and counting; the smaller margin on WSP is consistent with numerical-precision limits and error accumulation in long weighted sums dominating the final RMSE.

*Table 4.* Accuracy (%) on scene-graph QA. $\Delta$ reports the change relative to each method's base LLM.

| Model | Acc. (%) | Δ Acc. vs. Base (%) |
|---|---|---|
| Mistral | 45.95 | – |
| GOFA (Mistral-based) | 34.06 | -11.89 |
| Qwen3-8B | 66.00 | – |
| Weaver (Qwen3-based) | 79.20 | +13.20 |

**SceneGraph QA.** On SceneGraph QA, answers are free-form strings rather than fixed labels. Following standard

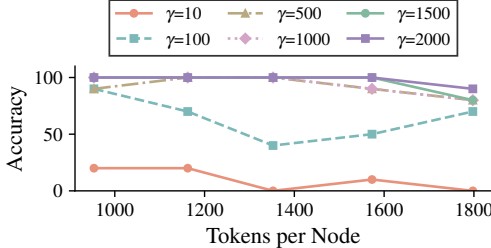

*Figure 6.* Impact of Local Prioritization $\gamma$ (Eq. (13)) on effective context length in the needle-in-a-haystack task.

LLM-as-judge protocols, we use GPT-5.4 to determine whether each prediction semantically matches the ground truth; the judge prompt is provided in Appendix C.3. As shown in Table 4, Weaver improves over its Qwen3-8B base by $+13.20$ points, whereas GOFA degrades its Mistral base by $-11.89$. This contrast suggests that Weaver injects structural signal without disrupting the base model's linguistic capability.

## 5.5. Computational Efficiency

To answer **RQ4**, we profile Weaver's inference latency and peak memory on a single A100-80GB GPU as input scales from 10K to 100K chars (Figure 5). Both metrics exhibit predictable scaling: latency grows from 1.04s to 35.39s following the expected $O(L^2)$ attention complexity, while memory scales linearly from 17.8 GB to 35.7 GB ($R^2 > 0.99$). The linear memory growth confirms that our FLEXATTENTION-based implementation effectively exploits Graph-Causal Mask sparsity. At 100K chars, Weaver consumes under 45% of GPU capacity, leaving headroom for larger graphs.

## 5.6. Ablations

**Max Node Context.** We analyze the Local Prioritization hyperparameter $\gamma$ (§3.2.3), which regulates attention bias toward intra-block information. Using a "needle-in-a-haystack" test on an arXiv-based ring graph, we append a unique marker to the end of a target node's text. Since the ring topology yields identical structural encodings, the model relies heavily on local prioritization to distinguish the target node. Figure 6 demonstrates that effective context length scales with $\gamma$. While low values ($\gamma \leq 100$) cause accuracy to collapse beyond 1,000 tokens due to positional collision, increasing $\gamma$ to 2,000 sustains near-perfect retrieval up to 1,800 tokens. This confirms that a strong local prior is essential for preventing the node representation from degenerating into a *bag-of-words*.

**Sequential *vs.* Reindexing.** We investigate the contribution of canonical reindexing and local prioritization to the model's robustness against node permutation using the Monochromatic Subgraph task (Table 5). The ablated model is highly sensitive to input order, with accuracy collapsing

*Table 5.* Monochromatic Subgraph (MS) Test with 200 samples under Node Permutation.

| | w/o (reindexing, local priority) | | w/ (reindexing, local priority) | |
|---|---|---|---|---|
| Metric | Original | Permuted | Original | Permuted |
| Accuracy (%) | 16.50 | 6.00 | 32.00 | 31.00 |
| RMSE | 4.87 | 5.19 | 4.57 | 4.55 |

from 16.5% to 6.0% under random permutation. In contrast, the full Weaver framework maintains superior and stable performance (32% vs. 31%); we attribute this marginal discrepancy primarily to the inherent stochasticity of the LLM sampling procedure rather than structural sensitivity. This confirms that our design effectively decouples reasoning from linear dependencies, ensuring the model relies on intrinsic graph topology.

## 6. Limitations and Future Work

**Scalability and Efficiency.** Weaver prioritizes reasoning expressiveness by modeling full graph semantics within the decoder, avoiding the information bottleneck of encoder-based pooling. This results in sequence length $L$ proportional to total graph text size and $\mathcal{O}(L^2)$ attention complexity. However, our FLEXATTENTION-based implementation (Dong et al., 2024) exploits Graph-Causal Mask sparsity, yielding linear memory scaling (Figure 5b). The framework permits flexible efficiency—spanning from global reasoning down to neighbor-only visibility—though further sparsification strategies (e.g., $k$-hop neighborhoods) remain future work.

**Modeling Heterophily.** On heterophilic benchmarks, Weaver significantly outperforms existing generative GLMs but does not yet match state-of-the-art specialized discriminative models (Table 2). We hypothesize this gap stems from the distribution of our alignment corpus, which relies primarily on homophilic graph-text pairs. Bridging this gap is a key direction for future work; we aim to curate a comprehensive training dataset covering diverse spectral properties and investigate adaptive prompting strategies to guide the model in generating context-sensitive embeddings for heterophilic structures.

## 7. Conclusion

We introduce a graph-over-tokens mechanism to decoder-only language models, allowing them to perform native topological reasoning without the information bottleneck of auxiliary encoders. When incorporated into a standard Transformer architecture, Weaver achieves state-of-the-art results on zero-shot transfer and algorithmic reasoning benchmarks, where it matches or exceeds the performance of specialized encoder-based GLMs and proprietary models like GPT-4o. Our results suggest that Weaver is a strong candidate to serve as a general backbone for unified graph-language intelligence.

## Acknowledgements

The work was supported in part by the Guangdong S&T Programme (Grant No. 2024B0101030002), the Basic Research Project No. HZQB-KCZYZ-2021067 of Hetao Shenzhen-HK S&T Cooperation Zone, the National Key Research and Development Program of China (Grant No. 2024YFB2907000), the National Natural Science Foundation of China (Grant No. 62293482 and Grant No. 62471423), the Shenzhen Science and Technology Program (Grant No. JCYJ20241202124021028 and Grant No. JCYJ20230807114204010), the Guangdong Talents Program (Grant No. 2024TQ08X346), the Shenzhen Outstanding Talents Training Fund 202002, the Guangdong Provincial Key Laboratory of Future Networks of Intelligence (Grant No. 2022B1212010001) and the Shenzhen Key Laboratory of Big Data and Artificial Intelligence (Grant No. SYSPG20241211173853027).

## Impact Statement

This paper presents Weaver, a framework that integrates graph structural inductive biases directly into the attention mechanism of Large Language Models (LLMs) without auxiliary encoders. Weaver has potential positive applications in areas such as scientific literature analysis, knowledge graph maintenance, and recommendation systems. As with any method leveraging pre-trained LLMs and relational data, standard considerations regarding data privacy and the potential inheritance of biases from the base model apply. We do not foresee immediate negative societal consequences beyond these general challenges inherent to the field.

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

# A. Comparing Weaver with other GLMs

*Table 6.* Comparison of design elements and capabilities across generative graph-language models.

| Method | External Encoders | Full Granularity | Conversation Support | Arbitrary TAG Support | Message Passing | Structural Inductive Bias |
|---|---|---|---|---|---|---|
| LINEARIZATION (Ye et al., 2024) | None | √ | √ | √ | – | × |
| GRAPHLLM (Chai et al., 2023) | E + GNN + P | × | √ | × | Global | √ |
| LLAGA (Chen et al., 2024) | E + T + P | × | √ | × | Global | √ |
| GOFA (Kong et al., 2025) | E + GNN + P | × | √ | √ | Local | √ |
| WEAVER (ours) | None | √ | √ | √ | Global | √ |

[†]E: Text Encoder, **GNN**: Graph Neural Network or Graph Transformer, **T**: Template, **P**: Multimodal Projector.

Table 6 summarizes how Weaver relates to representative generative Graph Language Models (GLMs). We group prior work by architectural paradigm and discuss the design choices that distinguish Weaver from each.

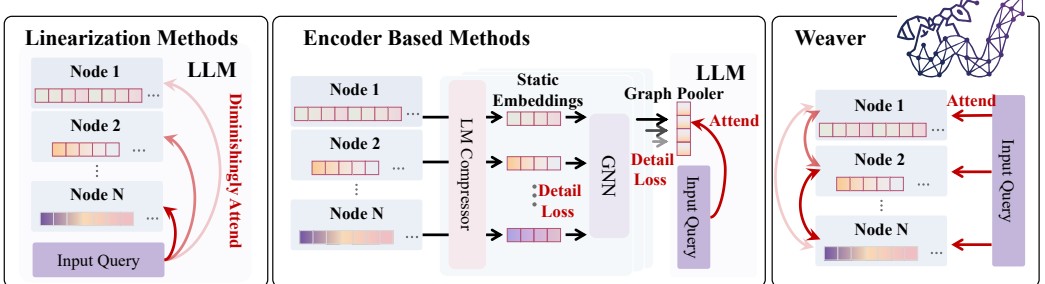

*Figure 7.* Architectural comparison between Weaver and prior GLM paradigms. **Left (Linearization):** node texts are concatenated into a single sequence and consumed by the LLM through standard causal attention. The query can attend to all node tokens, but attention weights diminish with sequential distance, and graph topology is conveyed only implicitly through token order. **Middle (Encoder-Based):** an external module (e.g., an LM compressor followed by a GNN and a graph pooler) encodes nodes into a small set of static embeddings before they reach the LLM. This pipeline injects structural inductive bias but compresses node content into fixed-size vectors, causing detail loss for long or fine-grained textual attributes. **Right (Weaver):** node texts are kept as full token blocks and fed directly to the LLM, while attention is reparameterized so that the input query attends to each node according to its structural position in the graph. Weaver thus retains token-level granularity as in linearization while incorporating topology-aware attention as in encoder-based methods, without an auxiliary encoder or projector.

**Versus Linearization Approaches (Figure 7, left).** Methods such as LINEARIZATION (Ye et al., 2024) flatten graph elements into a token sequence and rely on the LLM's pre-trained capacity to infer structure from textual descriptions (e.g., adjacency lists). This preserves *full granularity*, since every token remains accessible to attention, but provides no *structural inductive bias*: the model must recover topology from a 1D ordering imposed by causal masking and linear positional encodings. Performance on multi-hop reasoning tends to degrade as graphs grow. Weaver (Figure 7, right) retains the granularity of linearization and additionally injects structural information through GoRoPE, so that attention can directly use topological distances rather than only sequential position.

**Versus Encoder-Enhanced Models (Figure 7, middle).** A common GLM design, including GRAPHLLM (Chai et al., 2023), LLAGA (Chen et al., 2024), and GOFA (Kong et al., 2025), couples an external graph encoder (a GNN or Graph Transformer) to the LLM via a multimodal projector. We highlight three consequences of this design:

- **Information bottleneck.** Node features or subgraphs are pooled into fixed-size embeddings before being passed to the LLM, which compresses structural signals into a small set of vectors (Alon & Yahav, 2021). Fine-grained semantic detail in long textual attributes or edge descriptions is difficult to recover downstream. Weaver instead operates directly on tokens, so structural reasoning and semantic access share the same representation.
- **Arbitrary TAG support.** Handling edge attributes or heterogeneous node types in encoder-based methods often requires specialized components such as type-specific projection heads. In Weaver, edges and nodes are represented uniformly as token blocks with distinct structural coordinates, which keeps the backbone unchanged.
- **Message-passing range.** GNN encoders typically perform *local* aggregation over immediate neighbors, with the receptive field controlled by depth. Weaver allows attention between any pair of tokens, modulated by their structural distance, which lets the model weight local and long-range dependencies as required by the task.

**Versus Structure-Guided Efficient Attention.** Recent efficient attention mechanisms such as GRAPH-KV (Wang et al., 2025b) and Block-Attention (Ma et al., 2025) also exploit structural dependencies, but with a different objective: *context reduction*, i.e., pruning the KV cache to lower the cost of long sequences. While they share the technical ingredient of structured attention with Weaver, they target the selection of a compact context subset rather than reasoning over the graph structure. GRAPH-KV, in particular, is designed for Directed Acyclic Graphs (DAGs) and does not directly apply to the cyclic topologies that Weaver supports. It is also evaluated primarily on single-hop retrieval, and the authors report limited gains when extending to multi-hop settings (Wang et al., 2025b). Weaver instead encodes structural distance to support multi-hop message passing over general graphs while retaining the full semantic context.

In summary, Weaver combines a structural inductive bias with token-level access to graph content within a single decoder-only model, rather than through an auxiliary encoder or projector. This is consistent with our hypothesis that the limiting factor for LLMs on graph tasks is the 1D sequential bias of standard attention, not model capacity, and that reparameterizing attention visibility and positional structure is sufficient to support joint semantic-structural reasoning.

## B. Extended Discussion of Weaver

### B.1. Structural Scale and Aliasing

The rule-of-thumb for $s$ (Eq. 12) is derived from the need to avoid structural *wrap-around* (aliasing). If the induced phase shift $s\Delta\Psi\theta_{\min}$ exceeds $2\pi$, the position encoding becomes periodic and ambiguous. By setting $s \approx L_{\max}/\|\Delta\Psi\|_{\max}$, we map the largest structural discrepancy in the graph (e.g., the spectral diameter of the LapPE embedding) to the longest linear distance the pretrained model can represent. Intuitively, this aligns the dynamic range of the graph spectral coordinates (which often correlate with diffusion distance or effective resistance) with the backbone LLM's long-range positional capacity.

### B.2. Expanded Form of GoRoPE.

**Recall from 1-D Rotary Positional Encoding.** Rotary Positional Encoding (RoPE) (Su et al., 2021) applies coordinate-dependent rotations to token representations, enabling position-aware attention without absolute position embeddings. For a token at position $m$, RoPE transforms the word embedding $x_m \in \mathbb{R}^d$ into query and key vectors via

$$f_{\boldsymbol{W}_{\{q,k\}},\Theta}(x_m, m) = \boldsymbol{R}^d_{\Theta,m}\boldsymbol{W}_{\{q,k\}}\boldsymbol{x}_m, \tag{14}$$

where $\boldsymbol{W}_{\{q,k\}}$ are learnable projection matrices and $\boldsymbol{R}^d_{\Theta,m}$ is a block-diagonal rotation matrix:

$$\boldsymbol{R}^d_{\Theta,m} = \begin{pmatrix} \cos m\theta_1 & -\sin m\theta_1 & 0 & 0 & \cdots & 0 & 0 \\ \sin m\theta_1 & \cos m\theta_1 & 0 & 0 & \cdots & 0 & 0 \\ 0 & 0 & \cos m\theta_2 & -\sin m\theta_2 & \cdots & 0 & 0 \\ 0 & 0 & \sin m\theta_2 & \cos m\theta_2 & \cdots & 0 & 0 \\ \vdots & \vdots & \vdots & \vdots & \ddots & \vdots & \vdots \\ 0 & 0 & 0 & 0 & \cdots & \cos m\theta_{d/2} & -\sin m\theta_{d/2} \\ 0 & 0 & 0 & 0 & \cdots & \sin m\theta_{d/2} & \cos m\theta_{d/2} \end{pmatrix}, \tag{15}$$

with pre-defined frequency parameters $\Theta = \{\theta_i = 1000000^{-2(i-1)/d}, i \in [1, 2, \ldots, d/2]\}$ (In Qwen3). The elegance of RoPE lies in its translational invariance: the inner product of queries and keys depends only on relative positions,

$$\boldsymbol{q}^\top_m \boldsymbol{k}_n = (\boldsymbol{R}^d_{\Theta,m}\boldsymbol{W}_q x_m)^\top (\boldsymbol{R}^d_{\Theta,n}\boldsymbol{W}_k x_n) = \boldsymbol{x}^\top_m \boldsymbol{W}^\top_q \boldsymbol{R}^d_{\Theta,n-m}\boldsymbol{W}_k \boldsymbol{x}_n. \tag{16}$$

Taking advantage of the sparsity of $\boldsymbol{R}^d_{\Theta,m}$ in Equation (15), RoPE is always implemented with a computationally efficient

realization of the multiplication $\boldsymbol{R}_{\Theta,m}^d \boldsymbol{x}$ for $\boldsymbol{x} \in \mathbb{R}^d$ is:

$$
\boldsymbol{R}_{\Theta,m}^d \boldsymbol{x} = \begin{pmatrix} x_1 \\ x_2 \\ x_3 \\ x_4 \\ \vdots \\ x_{d-1} \\ x_d \end{pmatrix} \otimes \begin{pmatrix} \cos m\theta_1 \\ \cos m\theta_1 \\ \cos m\theta_2 \\ \cos m\theta_2 \\ \vdots \\ \cos m\theta_{d/2} \\ \cos m\theta_{d/2} \end{pmatrix} + \begin{pmatrix} -x_2 \\ x_1 \\ -x_4 \\ x_3 \\ \vdots \\ -x_d \\ x_{d-1} \end{pmatrix} \otimes \begin{pmatrix} \sin m\theta_1 \\ \sin m\theta_1 \\ \sin m\theta_2 \\ \sin m\theta_2 \\ \vdots \\ \sin m\theta_{d/2} \\ \sin m\theta_{d/2} \end{pmatrix},
\tag{17}
$$

where $\otimes$ denotes element-wise (Hadamard) product. This formulation reduces the complexity from $O(d^2)$ matrix multiplication to $O(d)$ element-wise operations, requiring only pre-computed sine and cosine values for each position and frequency pair. This is equivalent to Eq. (3).

**GoRoPE Formulation.** Similarly, the full rotation matrix from Eq. (9) can be written as

$$
\boldsymbol{R}_{\Theta,\boldsymbol{r}_u,m}^d = \begin{pmatrix} \cos(m+s\psi_{u,1})\theta_1 & -\sin(m+s\psi_{u,1})\theta_1 & 0 & 0 & \cdots & 0 & 0 \\ \sin(m+s\psi_{u,1})\theta_1 & \cos(m+s\psi_{u,1})\theta_1 & 0 & 0 & \cdots & 0 & 0 \\ 0 & 0 & \cos(m+s\psi_{u,2})\theta_2 & -\sin(m+s\psi_{u,2})\theta_2 & \cdots & 0 & 0 \\ 0 & 0 & \sin(m+s\psi_{u,2})\theta_2 & \cos(m+s\psi_{u,2})\theta_2 & \cdots & 0 & 0 \\ \vdots & \vdots & \vdots & \vdots & \ddots & \vdots & \vdots \\ 0 & 0 & 0 & 0 & \cdots & \cos(m+s\psi_{u,p})\theta_{d/2} & -\sin(m+s\psi_{u,p})\theta_{d/2} \\ 0 & 0 & 0 & 0 & \cdots & \sin(m+s\psi_{u,p})\theta_{d/2} & \cos(m+s\psi_{u,p})\theta_{d/2} \end{pmatrix},
\tag{18}
$$

which is a MRoPE-style multimodal expansion for graphs that takes exactly the same $\mathcal{O}(d)$ complexity.

### B.3. Design Choices for Local Prioritization

We consider two main approaches for biasing attention toward local blocks: *additive penalties* and *multiplicative decay*.

**Additive biases.** Our method (Eq. 13) subtracts a fixed offset $\gamma\beta_h$ from cross-block attention logits. Because this bias can drive logits to arbitrarily large negative values, the post-softmax attention weights of distant tokens approach zero exponentially. This provides strong locality control when $\gamma$ is large, while remaining negligible when content-based attention is dominant.

**Multiplicative decay.** Mechanisms such as xPos (Sun et al., 2023) scale attention logits by a factor $\exp(-\lambda d)$ based on relative distance $d$. However, multiplicative decay can only reduce logits toward zero, not below it. Consequently, softmax normalization still distributes probability mass across all positions, diluting rather than suppressing distant tokens. For instance, scaling a logit from $5.0$ to $0.5$ reduces its contribution but does not eliminate it. In contrast, subtracting $10.0$ to produce $-5.0$ yields near-zero post-softmax probability.

## C. Additional Experimental Details

### C.1. Benchmarks and Dataset Statistics

We evaluate Weaver on 15 real-world text-attributed graph benchmarks and 4 synthetic graph reasoning tasks, grouped into three categories: homophilic graphs, heterophilic graphs, and graph question answering (Graph QA). They cover three task types: node classification (NC), link prediction (LP), and Graph QA. Each node is associated with raw text rather than precomputed feature vectors. We obtain the raw graphs and node texts from two public repositories, LLMBP (Wang et al., 2025a)[2] and TAGLAS (Feng et al., 2024)[3], and adapt them to a unified format for our experiments. LLMBP supplies Citeseer, History, Children, Sportsfit, and the four WebKB graphs; TAGLAS supplies Cora, Pubmed, WikiCS, ExplaGraphs, and SceneGraphs.

**Homophilic benchmarks.** We use seven homophilic graphs spanning three domains: citation networks (Cora, Citeseer, Pubmed), Amazon co-purchase graphs (History, Children, Sportsfit), and a Wikipedia hyperlink graph (WikiCS). Node texts are titles and abstracts for citation graphs, product titles and descriptions for e-commerce graphs, and entry titles with article

---

[2] https://github.com/Graph-COM/LLM_BP
[3] https://github.com/JiaruiFeng/TAGLAS

*Table 7.* Dataset statistics and task specifications. "Hom." denotes the homophily ratio (Zhu et al., 2020)

| Dataset | Avg. #Nodes | Avg. #Edges | #Graphs | #Classes | Hom. | Task | Domain | Text Attributes |
|---|---|---|---|---|---|---|---|---|
| *Homophilic graphs* | | | | | | | | |
| Cora (McCallum et al., 2000) | 2,708 | 10,556 | 1 | 7 | 0.809 | NC, LP | Co-citation | Title + Abstract |
| Pubmed (Sen et al., 2008) | 19,717 | 88,648 | 1 | 3 | 0.792 | NC, LP | Co-citation | Title + Abstract |
| Arxiv (Hu et al., 2020) | 169,343 | 1,166,243 | 1 | 40 | 0.658 | NC, LP | citation | Title + Abstract |
| Products (Hu et al., 2020) | 54,025 | 144,638 | 1 | 47 | 0.812 | NC, LP | Co-purchase | Title / Description |
| Citeseer (Giles et al., 1998) | 3,186 | 8,450 | 1 | 6 | 0.764 | NC | Citation | Title + Abstract |
| History (Ni et al., 2019) | 41,551 | 150,318 | 1 | 12 | 0.662 | NC | E-commerce | Title / Description |
| Children (Ni et al., 2019) | 76,875 | 523,250 | 1 | 24 | 0.464 | NC | E-commerce | Title / Description |
| Sportsfit (Ni et al., 2019) | 173,055 | 3,020,134 | 1 | 13 | 0.900 | NC | E-commerce | Title / Description |
| WikiCS (Mernyei & Cangea, 2020) | 11,701 | 431,726 | 1 | 10 | 0.678 | NC | Web/Knowledge | Entry + Content |
| *Heterophilic graphs (WebKB)* | | | | | | | | |
| Cornell (Craven et al., 1998) | 191 | 292 | 1 | 5 | 0.115 | NC | Webpage | HTML content |
| Texas (Craven et al., 1998) | 187 | 310 | 1 | 5 | 0.067 | NC | Webpage | HTML content |
| Wisconsin (Craven et al., 1998) | 265 | 515 | 1 | 5 | 0.152 | NC | Webpage | HTML content |
| Washington (Craven et al., 1998) | 229 | 394 | 1 | 5 | 0.149 | NC | Webpage | HTML content |
| *Graph QA* | | | | | | | | |
| ExplaGraphs (Saha et al., 2021) | 5.17 | 4.25 | 2,766 | 2 | – | Graph QA | Commonsense | Concept |
| SceneGraphs (Hudson & Manning, 2019) | 19.13 | 68.44 | 100,000 | – | – | Graph QA | Scene graph | Object |
| Structural Reasoning (synthetic) | 30.05 | 304.44 | 300,000 | – | – | Graph QA | Synthetic | Node attr. / Edge weights |

content for WikiCS. We evaluate NC on all seven datasets and additionally evaluate LP on Cora and Pubmed.

**Heterophilic benchmarks.** We evaluate on four WebKB webpage graphs (Cornell, Texas, Wisconsin, and Washington), where edges correspond to hyperlinks between pages and node texts are the raw webpage contents. The task is 5-class node classification over *student*, *faculty*, *staff*, *project*, and *course*. Linked nodes frequently belong to different classes, making these graphs strongly heterophilic.

**Graph QA benchmarks.** We use two real-world Graph QA datasets and four synthetic diagnostic tasks. ExplaGraphs is a commonsense reasoning benchmark formulated as binary classification over argument pairs grounded in a concept graph, with concept text on nodes. SceneGraph dataset consists of image-derived scene graphs and requires open-ended question answering conditioned on object nodes and their relations. To further probe whether Weaver explicitly models and exploits graph topology, we additionally construct four synthetic tasks: Shortest Path (SP), Weighted Shortest Path (WSP), Common Neighbors (CN), and Monochromatic Subgraph (MS). Task definitions, data generation, and evaluation metrics are detailed in Appendix C.2.

### C.2. Synthetic Structural Reasoning Tasks

To assess whether Weaver explicitly encodes and exploits graph topology rather than relying on semantic cues, we construct four synthetic graph algorithmic tasks that require multi-hop reasoning over structure. The correct answer is determined solely by the graph topology (and, for MS, additionally by node colors), so node text provides no shortcut.

**Tasks.** Each task is defined on a graph $G = (V, E)$ with $|V| = n$, and $\Gamma(u)$ denotes the one-hop neighborhood of $u$.

- **Shortest Path (SP).** Given an unweighted graph and a source/target pair $s, t \in V$, predict (i) the shortest-path length $d(s, t)$ in edges and (ii) a valid shortest path from $s$ to $t$.
- **Weighted Shortest Path (WSP).** Given a graph with positive integer edge weights $\{w_e\}_{e \in E}$ and a pair $s, t$, predict (i) the minimum total weight $d_w(s, t)$ and (ii) a path from $s$ to $t$ achieving this minimum.
- **Common Neighbors (CN).** Given a pair $u, v \in V$, predict (i) the size $|\Gamma(u) \cap \Gamma(v)|$ and (ii) the set $\Gamma(u) \cap \Gamma(v)$.
- **Monochromatic Subgraph (MS).** Given a node-colored graph with colors $c_u \in \{0, \ldots, k-1\}$ and a target color $c^\star$, predict the node set of the largest connected component in the subgraph induced by $\{u \in V : c_u = c^\star\}$.

**Data generation.** For each task we sample Erdős–Rényi random graphs $G(n, p)$ with graph sizes $n \in \{5, 10, 15, \ldots, 50\}$ and edge probability $p \le 0.3$, resampling until the graph is connected. For each $n$ we generate 500 test instances per task. For WSP, edge weights are drawn i.i.d. uniformly from $\{1, \ldots, 1000\}$. For MS, node colors are drawn i.i.d. uniformly from $k = 3$ colors. Source/target pairs (SP, WSP) and node pairs (CN) are sampled uniformly at random from distinct nodes in $V$.

## C.3. Judge Prompt for SceneGraph QA

Answers in SceneGraph QA are free-form strings, and exact match tends to undercount correct predictions that simply use different wording from the reference (e.g., `"a man"` vs. `"person"`). To avoid this, we score predictions with an LLM judge using the prompt shown in Figure 8.

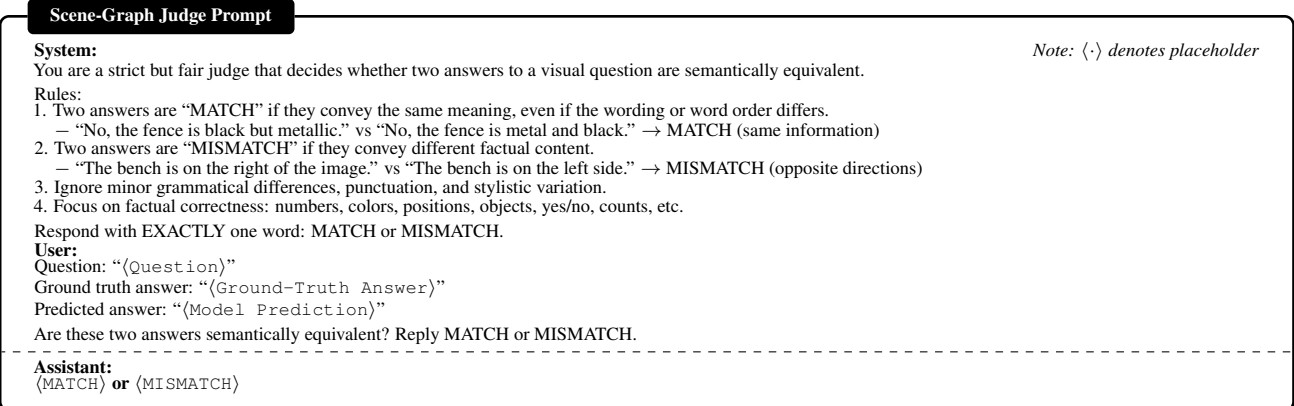

**Scene-Graph Judge Prompt**

**System:**                                                                                                    *Note: ⟨·⟩ denotes placeholder*
You are a strict but fair judge that decides whether two answers to a visual question are semantically equivalent.
Rules:
1. Two answers are "MATCH" if they convey the same meaning, even if the wording or word order differs.
    — "No, the fence is black but metallic." vs "No, the fence is metal and black." → MATCH (same information)
2. Two answers are "MISMATCH" if they convey different factual content.
    — "The bench is on the right of the image." vs "The bench is on the left side." → MISMATCH (opposite directions)
3. Ignore minor grammatical differences, punctuation, and stylistic variation.
4. Focus on factual correctness: numbers, colors, positions, objects, yes/no, counts, etc.

Respond with EXACTLY one word: MATCH or MISMATCH.
**User:**
Question: "⟨Question⟩"
Ground truth answer: "⟨Ground-Truth Answer⟩"
Predicted answer: "⟨Model Prediction⟩"

Are these two answers semantically equivalent? Reply MATCH or MISMATCH.
- - - - - - - - - - - - - - - - - - - - - - - - - - - - - - - - - - - - - - - - - - - - - - - - - - - - - - -
**Assistant:**
⟨MATCH⟩ **or** ⟨MISMATCH⟩

*Figure 8.* Judge prompt used for Scene-Graph QA evaluation. The judge receives the ground-truth answer and the model prediction, and emits a single token verdict. We then parse the response with a regular expression.

## C.4. Baselines

We compare WEAVER with two groups of baselines: *specialized* methods that obtain label predictions through embedding and aggregation pipelines, and *generative* methods that formulate TAG inference as conditional text generation. For comparability, we adopt the standardized evaluation protocols of Wang et al. (2025a) and Kong et al. (2025), which specify the raw text attributes, data splits, label spaces, and evaluation metrics used in our experiments (Appendix C). When a baseline has been evaluated under the same protocol, we report the published result rather than re-running the method. This avoids introducing additional variation from implementation choices or hyperparameter retuning and preserves the original experimental setup of each method. Unless otherwise noted, baseline results are taken from Wang et al. (2025a); GOFA results are taken from Kong et al. (2025).

**Specialized methods.** SBERT (Reimers & Gurevych, 2019) and ROBERTA (Zhuang et al., 2021) are text-only encoder baselines that classify nodes using embeddings of their text attributes. SBERT+NA (Yang et al., 2024) is the training-free Neighborhood Aggregation (NA) baseline, which incorporates graph structure by aggregating neighborhood text embeddings. DGI (Veličković et al., 2019) and GRAPHMAE (Hou et al., 2022) are self-supervised GNN representation learning baselines, evaluated in the zero-shot setting. UNIGLM (Fang et al., 2025) and ZEROG (Li et al., 2024) tune LM encoders for cross-dataset generalization on TAGs. OFA (Liu et al., 2024) is a multi-task graph foundation model baseline. TEXT-EMBEDDING-3-LARGE (OpenAI, 2024b) and LLM2VEC (BehnamGhader et al., 2024) are text embedding baselines. LLM-BP (Wang et al., 2025a) is a training-free belief-propagation framework that combines task-adaptive embeddings with belief propagation using an LLM-estimated homophily parameter.

**Generative methods.** GPT-3.5-TURBO (OpenAI, 2024) and GPT-4O (OpenAI, 2024a) are vanilla LLM baselines that directly classify nodes from raw node texts without incorporating graph structure. We evaluate QWEN3-8B-FT under the same text-only protocol: it fine-tunes Qwen3 on raw node texts only and excludes graph-structural information, serving as an ablation of the base model used by WEAVER. GRAPHGPT (Tang et al., 2024) is a graph instruction-tuning method. LLAGA (Chen et al., 2024) is an LLM-based framework that converts a node-centered subgraph into a structure-aware node sequence using templates such as Neighborhood Detail and Hop-Field Overview, and learns a projector from node representations to the LLM token embedding space while keeping the LLM frozen. GOFA (Kong et al., 2025) is a generative graph-language model that interleaves GNN layers with a frozen LLM and is trained with graph completion, multiple pre-training tasks, and instruction fine-tuning.

## C.5. Experimental Hyperparameters

Table 8 summarizes the hyperparameters used for WEAVER's alignment and downstream fine-tuning stages, respectively (§3.3). Unless otherwise specified, all experiments use Qwen3-8B[4] as the backbone and are run on a server with $8\times$A100-80GB GPUs.

*Table 8.* Hyperparameters for Weaver-Zero alignment and downstream fine-tuning.

| Hyperparameter | Alignment | Fine-tuning |
|---|---|---|
| Learning rate | $5 \times 10^{-5}$ | $1 \times 10^{-5}$ |
| Training steps | 9,340 | 9,000 |
| Batch size | 32 | 16 |
| Warmup steps | 500 | 200 |
| Weight decay | 0 | 0 |
| LoRA rank ($r$) | 32 | 8 |
| LoRA alpha | 64 | 8 |
| Random seed | 42 | 42 |
| $s$ | 10k | 10k |
| $\gamma$ | 2k | 2k |

# D. Additional Experiments

## D.1. Zero-shot Performance vs. Label Set Size

*Table 9.* Zero-shot node classification accuracy (%) under varying label set sizes.

| | Cora | | Products | | | WikiCS | | ExplaGraphs |
|---|---|---|---|---|---|---|---|---|
| Label Set Size | 7 | 2 | 47 | 10 | 5 | 10 | 5 | 2 |
| OFA | 28.7 | 56.9 | 19.4 | 30.4 | 39.3 | 21.2 | 35.2 | 51.4 |
| UNIGRAPH | 69.5 | 89.7 | 38.5 | 66.1 | 75.7 | 43.5 | 60.2 | – |
| ZEROG | 64.2 | 87.8 | 31.2 | 51.2 | 71.3 | 31.3 | 48.3 | – |
| LLAGA | 51.9 | 62.7 | 23.1 | 34.2 | 39.7 | – | – | – |
| GOFA-T | 70.8 | 85.7 | 54.6 | 79.3 | 87.1 | **71.2** | 80.9 | 79.5 |
| Ours | **73.0** | **89.8** | **68.5** | **84.5** | **91.4** | 69.4 | **81.1** | **82.4** |

Table 9 reports zero-shot node classification accuracy across four datasets while varying the number of candidate labels presented at inference. We summarize three observations.

**Sensitivity to label set size.** Accuracy improves for all methods as the candidate set shrinks, consistent with reduced task difficulty. The magnitude of this effect varies: on Cora, OFA changes by 28.2 points between the 7-class and 2-class settings ($28.7 \rightarrow 56.9$), whereas Weaver changes by 16.8 points ($73.0 \rightarrow 89.8$). A similar pattern holds on Products and WikiCS, where Weaver and GOFA-T are less sensitive to label granularity than the remaining baselines.

**Behavior on large label spaces.** The gap between Weaver and the strongest baseline is largest on Products at the full 47-class setting (68.5 vs. 54.6 for GOFA-T, $+13.9$ points), and narrows as the candidate set is reduced ($+5.2$ at 10 classes, $+4.3$ at 5 classes). Products also has the lowest homophily among the evaluated datasets (0.21); we report this co-occurrence without claiming a causal relationship.

**Overall comparison.** Weaver attains the highest accuracy in 7 of 8 (dataset, label-set) configurations. The exception is WikiCS at 10 classes, where GOFA-T leads by 1.8 points (71.2 vs. 69.4); the two methods are within 0.2 points at 5 classes. On ExplaGraphs (binary), WEAVER reaches $82.4\%$ versus $79.5\%$ for GOFA-T; UNIGRAPH, ZEROG, and LLAGA do not report results on this dataset.

---

[4]https://huggingface.co/Qwen/Qwen3-8B

### D.2. Structural Reasoning across Graph Sizes

While the main text reports aggregate improvements over the Qwen3-8B backbone on the synthetic tasks, here we provide a per-task, per-size breakdown using metrics tailored to each output type.

**Metrics.** For each task we average the following metrics over the 500 instances at each graph size $n$:

- **Acc** (SP/WSP/CN): exact match of the scalar answer ($d(s,t)$, $d_w(s,t)$, or $|\Gamma(u) \cap \Gamma(v)|$).
- **Path Acc** (SP/WSP): fraction of instances for which the generated path is a valid shortest path from $s$ to $t$.
- **Node Acc** (CN): exact match between the predicted and ground-truth common-neighbor sets.
- **Node F1** (MS): F1 between the predicted node set and the ground-truth largest monochromatic component.

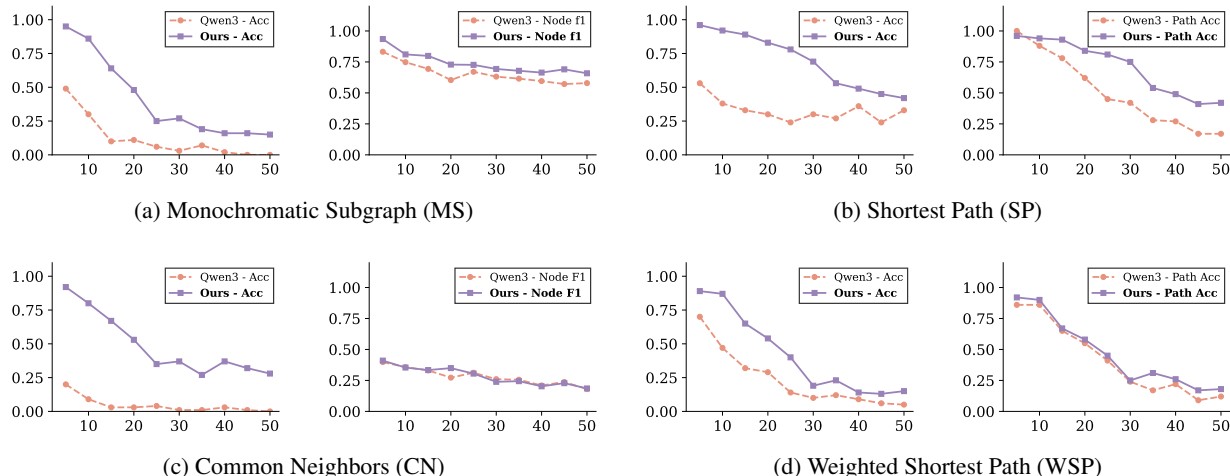

(a) Monochromatic Subgraph (MS)    (b) Shortest Path (SP)

(c) Common Neighbors (CN)    (d) Weighted Shortest Path (WSP)

*Figure 9.* Comparison between our model and the Qwen3 base model across four graph reasoning tasks. The $x$-axis denotes the number of nodes. For each task, the left plot shows accuracy (Acc) of the predicted numerical value, and the right plot shows task-specific metrics: Node F1 (MS), Path Acc (SP, WSP), and Node F1 (CN).

**Results.** Figure 9 compares Weaver with its Qwen3-8B backbone across the four tasks and graph sizes $n \in \{5, \ldots, 50\}$. Weaver outperforms Qwen3-8B on all tasks at nearly all $n$, and the gap generally widens with $n$, most visibly on CN and MS. On SP and WSP, Weaver maintains higher Path Acc at larger graph sizes, indicating more reliable recovery of valid shortest paths. Both models degrade as $n$ grows, but Weaver degrades more smoothly, while Qwen3-8B exhibits larger variance on CN and MS.

### D.3. Qualitative Case Studies

Below, we provide qualitative case studies to complement the quantitative results in §5. These examples examine Weaver's behavior in open-ended generative question answering over graph-structured inputs, where the model is required to produce graph-grounded free-form responses. Figure 10 presents representative success cases on the structural reasoning tasks. Figure 11 presents SceneGraph visual question answering examples over object-relation graphs. Finally, Table 10 reports representative failure cases of Weaver-Zero. The failure cases mainly involve incomplete enumeration of valid nodes or shortest paths, as well as arithmetic errors when aggregating edge weights.

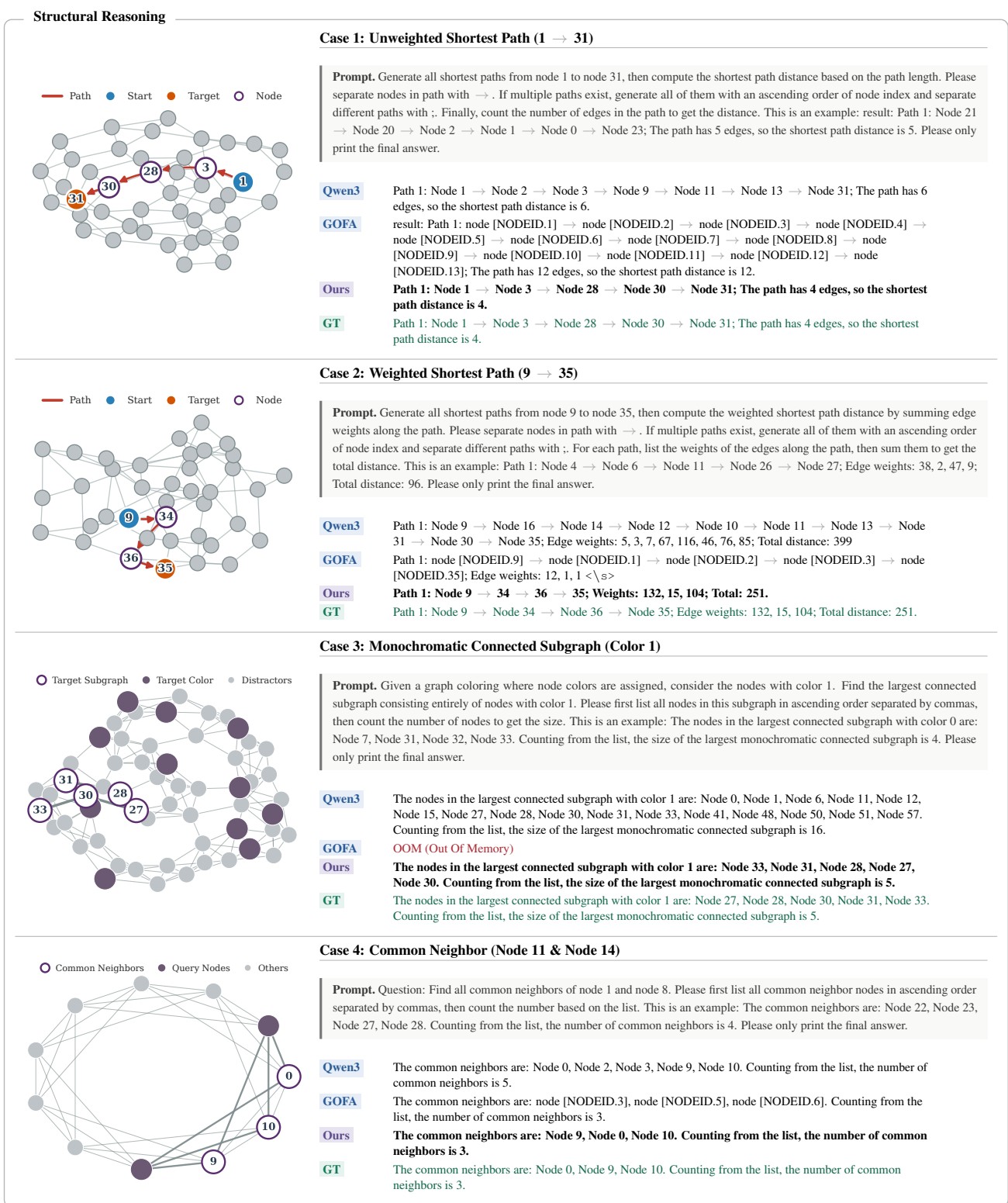

*Figure 10.* Qualitative comparison across four graph reasoning tasks. Green indicates correct answers matching ground truth (GT). In these examples, our method produces accurate results while baselines exhibit errors or failures.

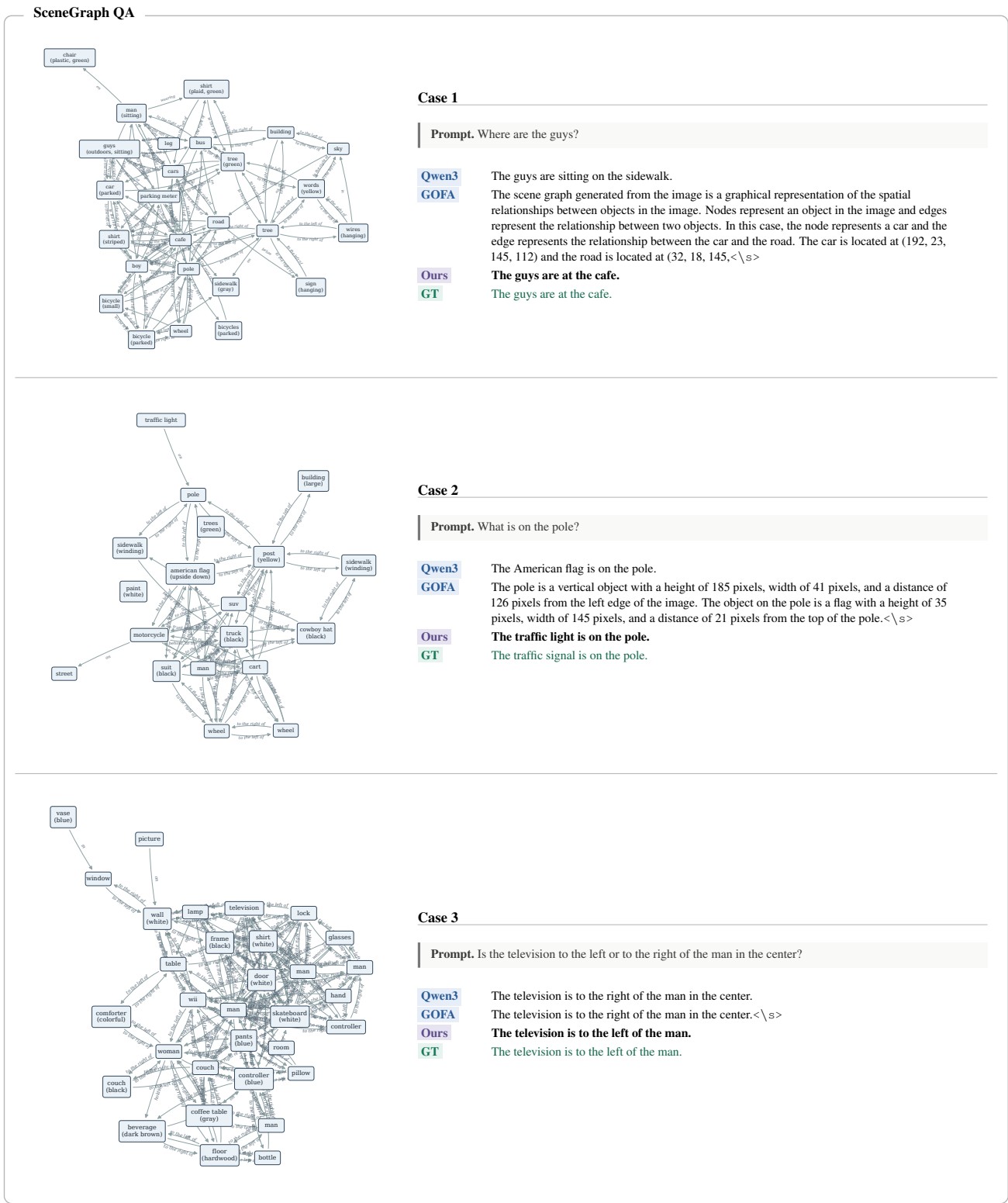

*Figure 11.* Qualitative evaluation on graph question answering with scene graphs. Green indicates answers matching the ground truth (GT). Across the examples, our method correctly reasons over object relationships and spatial layouts, while baseline models produce inaccurate or inconsistent responses.

*Table 10.* Case study: Error analysis across graph reasoning tasks

| Attribute | Details |
|---|---|
| Input | *50 nodes graph & CN prompt* |
| Error Type | Incomplete enumeration |
| Ground Truth | The common neighbors are: Node 13, Node 14, Node 15, Node 16. Counting from the list, the number of common neighbors is 4. |
| Model Prediction | The common neighbors are: Node 13. Counting from the list, the number of common neighbors is 1. |
| Input | *50 nodes graph & CN prompt* |
| Error Type | Incomplete enumeration |
| Ground Truth | The common neighbors are: Node 10, Node 15, Node 16, Node 17. Counting from the list, the number of common neighbors is 4. |
| Model Prediction | The common neighbors are: Node 15, Node 10. Counting from the list, the number of common neighbors is 2. |
| Input | *50 nodes graph & MS prompt (color 1)* |
| Error Type | Missing nodes |
| Ground Truth | The nodes in the largest connected subgraph with color 1 are: Node 4, Node 7, Node 13, Node 15, Node 16, Node 18, Node 22, Node 24, Node 27, Node 40, Node 43. Counting from the list, the size of the largest monochromatic connected subgraph is 11. |
| Model Prediction | The nodes in the largest connected subgraph with color 1 are: Node 18, Node 16, Node 11, Node 22, Node 27, Node 43, Node 40, Node 7, Node 1. Counting from the list, the size of the largest monochromatic connected subgraph is 9. |
| Input | *50 nodes graph & MS prompt (color 0)* |
| Error Type | Missing nodes |
| Ground Truth | The nodes in the largest connected subgraph with color 0 are: Node 9, Node 10, Node 11, Node 24, Node 26, Node 37, Node 39, Node 40, Node 41, Node 43. Counting from the list, the size of the largest monochromatic connected subgraph is 10. |
| Model Prediction | The nodes in the largest connected subgraph with color 0 are: Node 24, Node 26, Node 40, Node 41, Node 39, Node 11, Node 9, Node 10. Counting from the list, the size of the largest monochromatic connected subgraph is 8. |
| Input | *50 nodes graph & SP prompt* |
| Error Type | Incomplete path enumeration |
| Ground Truth | Path 1: Node 13 → Node 12 → Node 10; Path 2: Node 13 → Node 14 → Node 10; Path 3: Node 13 → Node 8 → Node 10; Path 4: Node 13 → Node 7 → Node 10; The path has 2 edges, so the shortest path distance is 2. |
| Model Prediction | Path 1: Node 13 → Node 12 → Node 10; The path has 2 edges, so the shortest path distance is 2. |
| Input | *50 nodes graph & SP prompt* |
| Error Type | Incomplete path enumeration |
| Ground Truth | Path 1: Node 9 → Node 11 → Node 17; Path 2: Node 9 → Node 13 → Node 17; Path 3: Node 9 → Node 14 → Node 17; Path 4: Node 9 → Node 20 → Node 17; The path has 2 edges, so the shortest path distance is 2. |
| Model Prediction | Path 1: Node 9 → Node 14 → Node 17; The path has 2 edges, so the shortest path distance is 2. |
| Input | *50 nodes graph & WSP prompt* |
| Error Type | Arithmetic error |
| Ground Truth | Path 1: Node 9 → Node 10 → Node 11 → Node 12 → Node 6 → Node 5 → Node 4; Edge weights: 95, 173, 56, 80, 43, 38; Total distance: 485. |
| Model Prediction | Path 1: Node 9 → Node 10 → Node 11 → Node 12 → Node 6 → Node 5 → Node 4; Edge weights: 95, 173, 56, 80, 43, 38; Total distance: 585. |
| Input | *50 nodes graph & WSP prompt* |
| Error Type | Arithmetic error |
| Ground Truth | Path 1: Node 16 → Node 17 → Node 18 → Node 19 → Node 0 → Node 1 → Node 2 → Node 3; Edge weights: 88, 118, 2, 148, 103, 136, 3; Total distance: 598. |
| Model Prediction | Path 1: Node 16 → Node 17 → Node 18 → Node 19 → Node 0 → Node 1 → Node 2 → Node 3; Edge weights: 88, 118, 2, 148, 103, 136, 3; Total distance: 600. |

