# OpenReview forum: "Weaving Graph over Tokens: Contextualizing Structured Sequences for LLMs"
_ICML.cc/2026/Conference — ICML 2026 regular_

### Official Review · Reviewer_J82j · 2026-03-09

**Soundness:** 3
**Presentation:** 3
**Significance:** 2
**Originality:** 3
**Overall Recommendation:** 4
**Confidence:** 3

**Summary:**

Overall, this article considers a central concept: integrating graph topology into the attention mechanism of decoder-only LLMs to enable structural reasoning over text-attributed graphs. The paper proposes Weaver, which modifies the attention mechanism of LLMs through Graph-Causal Masking and Graph-over-Tokens RoPE (GoRoPE) to encode graph topology directly in attention without relying on external graph encoders.

**Compliance With Llm Reviewing Policy:**

Affirmed.

**Final Justification:**

The author has addressed my concerns.

**Key Questions For Authors:**

See weakness.

**Limitations:**

yes

**Strengths And Weaknesses:**

Strength:

The paper proposes an elegant modification of the attention mechanism to incorporate graph structure directly into token-level reasoning. Integrating topology into positional encoding (GoRoPE) is conceptually simple and aligns well with the transformer attention framework.

The proposed method injects structural bias directly into the decoder-only architecture. This avoids the information bottleneck of encoder-based pipelines and provides a unified generative framework for graph-language tasks.

The paper evaluates the method across multiple benchmarks including node classification, link prediction, and graph reasoning tasks.

Weakness:

The main motivation of the method is to enable topology-aware reasoning in LLMs. However, the experimental results do not clearly demonstrate that the model effectively utilizes structural information in challenging settings. For example, the model still performs significantly worse than specialized approaches such as LLM-BP on heterophilic graphs. This raises the question of whether the observed gains mainly come from textual semantic signals rather than genuine structural reasoning. Therefore, I suggest to include the train-from-scrach LLMs architechtures, eg., Qwen3 architectures

Some important baselines for LLM-based graph learning are missing, such as LLM-BP and OFA. Including these methods would provide a more complete comparison with recent approaches that explicitly incorporate structural propagation in Table 3.

It’s confusing to apply the model. The authors introduce new token operations in the method part, which means the model need be trained from scratch. However, in 5.4, the author suggests that the model is based on Qwen3. This seems to be a fine-tuned method on the existing LMS.

Meanwhile, what is the formulation of structure reasoning tasks? Are they the same as the node classification tasks?

---

> ### Author Rebuttal · Authors · 2026-03-31
>
> Thank you for the thoughtful and constructive feedback, and for recognizing the elegance of our attention-based graph integration. We address your concerns below.
>
> >W1: The gains may come from textual semantics rather than structural reasoning...suggest to include train-from-scratch Qwen3.
>
> We thank the reviewer for the valuable suggestion. We agree that this comparison strengthens the paper. We will incorporate these experiments and analyses into the revised manuscript:
> - *Ablation on Qwen3.* Following the suggestion, we fine-tune Qwen3 on raw node texts without incorporating graph structures (Qwen3-ft) and compare against Weaver in the zero-shot classification setting (Table 2). Results show Weaver improves on 9/10 datasets, including challenging heterophilic benchmarks. Since the two models differ only in whether graph topology is injected, this improvement can be directly attributed to structural information rather than textual semantics.
> - *Spectral ablation on LapPE features.* To further isolate what type of structural signal Weaver exploits, we reduce the LapPE dimension from 16 to 8 (Weaver-8), discarding higher-frequency Laplacian eigenvectors that encode fine-grained local structural differences between nearby nodes. Results show Weaver-8 degrades sharply on heterophilic benchmarks while homophilic datasets remain relatively stable (see table below). If the gains were text-driven, truncating spectral dimensions would not produce this heterophily-specific degradation. This indicates that Weaver's performance on heterophilic graphs is sensitive to higher-frequency spectral features, suggesting that access to richer spectral positional encodings is an important factor in handling heterophily.
> - *Analysis of pure structural tasks.* We would also kindly refer the reviewer to Section 5.4, where Weaver is evaluated on purely structural tasks (Shortest Path, Common Neighbors, Monochromatic Subgraph, etc.) with textual shortcuts entirely absent. Weaver consistently outperforms the Qwen3-8B backbone across all graph sizes.
>
> Summary of extended results (for Table 2):
> |Model|Cora|Citeseer|History|Children|Sportsfit|WikiCS|Cornell|Texas|Wisconsin|Washington|
> |-|-|-|-|-|-|-|-|-|-|-|
> |Qwen3-ft|70.70|63.67|52.33|30.67|60.67|66.00|48.16|62.03|60.75|51.09|
> |Weaver|76.50|70.14|60.60|34.30|63.60|69.80|69.40|60.90|62.50|52.50|
> |Weaver-8|74.00|64.33|57.00|31.67|61.33|68.33|53.90|49.73|45.04|41.20|
>
> *Regarding the gap with LLM-BP on heterophilic graphs* (acknowledged in our Limitations): LLM-BP uses specialized, task-specific label propagation, whereas Weaver is a unified generative framework that consistently outperforms LLM-BP on homophilic graphs and narrows the gap on heterophilic ones.
>
> >W2: LLM-BP and OFA missing in Table 3.
>
> Currently, both LLM-BP and OFA are included in our zero-shot evaluation (Table 2). We have added their available reported results to Table 3 below for a more complete comparison:
>
> |Method|Paradigm|Pubmed (NC)|Arxiv (NC)|Pubmed (LP)|Arxiv (LP)|
> |-|-|-|-|-|-|
> |OFA|Embedding|75.51|73.44|95.89|-|
> |LLM-BP|Embedding|75.55|-|97.90|-|
> |GOFA|Generative|83.83|74.77|93.97|85.41|
> |LLaGA|Generative|81.56|74.46|89.81|92.04|
> |Weaver|Generative|86.20|85.90|94.20|88.10|
>
> They were originally separated as Table 3 compares methods under the same generative paradigm. We will clarify this in the revised caption.
> >W3: The method introduce new token operations...means the model need be trained from scratch. However...model is based on Qwen3. This seems to be a fine-tuned method...
>
> We would like to clarify that introducing new attention patterns does not require training from scratch. Graph-Causal Masking, GoRoPE, and biasing are all based on standard attention operations used in established extrapolation techniques [1, 2, 3], which adapt pre-trained LLMs via fine-tuning. As detailed in Section 3.3, we adapt Qwen3-8B with LoRA for 1 epoch. We will make this explicit at the beginning of Section 3.
>
> >W4: What is the formulation of structure reasoning tasks? Are they the same as the node classification tasks?
>
> They are different tasks but share the same graph-conditioned autoregressive formulation in Weaver. Given graph $G$, instruction $x$, and output $y=(y_1,\dots,y_T)$, Weaver models: $p(y\mid G,x)=\prod_{t=1}^T p(y_t\mid y_{<t},G,x).$ The tasks differ only in what $x$ asks and what $y$ contains. For node classification, $x$ queries a node's label; for structural reasoning (Section 5.4), $x$ queries topological properties (shortest paths, common neighbors, monochromatic subgraphs) on synthetic graphs without meaningful textual features. We will add an appendix table summarizing each task's $(G, x, y)$.
>
> [1] Press et al. Train Short, Test Long: Attention with Linear Biases Enables Input Length Extrapolation. ICLR 2022.
>
> [2] Peng et al. YaRN: Efficient Context Window Extension of Large Language Models. ICLR 2024.
>
> [3] Ma et al. Block-Attention for Efficient Prefilling. ICLR 2025.

---

> > ### Author Rebuttal · Reviewer_J82j · 2026-04-01
> >
> > Thanks for your response, will raise the score to 4.

---

### Official Review · Reviewer_g7N7 · 2026-03-09

**Soundness:** 3
**Presentation:** 3
**Significance:** 3
**Originality:** 3
**Overall Recommendation:** 4
**Confidence:** 5

**Summary:**

# Summary

Thanks for submitting your work to ICML. Below are my comments.

In this work, the authors focus on improving the performance of graph language models by introducing an encoder-free framework (weaver), which extends the attention mechanism of LLMs based on the rotary positional encoding.

To materialize such a design, the authors propose three techniques, including (1) a mask mechanism, (2) a unified geometric encoder, and (3) a local information prioritization principle.

Experimental results show that Weaver is effective on homophilic graphs.

Due to the current limitations in both the experimental evaluation and the theoretical analysis, some revision work is still needed.

**Compliance With Llm Reviewing Policy:**

Affirmed.

**Final Justification:**

Thanks for the authors' detailed response. The additional experimental comparisons and the detailed analysis presented in the rebuttal have successfully resolved my concerns. I am keeping my positive rating and upgrading the Originality and Soundness scores from fair to good.

**Key Questions For Authors:**

# Questions:

(1) Why can the introduction of structural coordinates into RoPE improve performance? Is there any experimental and theoretical analysis that explains this effect?

(2) What exactly is the information bottleneck caused by external encoders? Is there any theoretical analysis that demonstrates that such an information bottleneck exists?

(3) What is the key technical factor that makes Weaver effective compared to GOFA? Is it simply the removal of the external encoder combined with the addition of positional encoding? Would adding an additional encoder to Weaver lead to a degradation in performance? Providing more detailed ablation studies would help better identify and clarify the advantages of Weaver.

**Limitations:**

yes

**Strengths And Weaknesses:**

# Strength

\+ **Significance**: This paper presents a new perspective showing that GLM performance can be enhanced without using an external encoder.

\+ **Presentation:** This paper is well-written.

\+ Experimental results demonstrate the superiority of the proposed model on homophilic graphs.

# Weakness

\- **Soundness:** The core technical contribution of this paper lies in extending the existing RoPE mechanism by incorporating structural coordinates. However, this extension appears somewhat incremental from an intuitive perspective. The paper lacks a theoretical analysis explaining why this extension is effective. In addition, there is no ablation study comparing different positional encoding strategies, which makes it difficult to clearly assess the benefit of the proposed design.

\- **Originality:** The core claim of this paper is that external encoders are unnecessary. However, the discussion supporting this claim is currently not sufficiently thorough. The authors claim that Weaver can avoid the information bottleneck. However, the concept of the information bottleneck is rather abstract in the current presentation and lacks a sufficiently formal explanation.

\-  Figure 1 contains a substantial amount of information. However, the textual description of the figure is rather limited. It is recommended to provide a more detailed caption with additional explanations of the components and key observations in Figure 1 to improve the readability of the paper. Besides, the experimental results do not report standard deviations.

Moreover, the paper lacks experiments that incorporate external encoders into Weaver to verify the claim that external encoders are indeed unnecessary.

Besides, the discussion comparing the architecture of Weaver with existing baselines is insufficient in the main text. Although this content is mentioned in the appendix, it would be beneficial to include a more concise and clear comparison in the main paper.

---

> ### Author Rebuttal · Authors · 2026-03-31
>
> Thank you for your constructive review and kind comments! We response to your comments below.
>
> >Q1/W1: Why can the introduction of structural coordinates into RoPE improve performance?
>
> Standard 1D RoPE assigns monotonically increasing scalar positions during sequence flattening, imposing a chain-graph prior on attention decay. This contradicts graph topology, where sequentially distant nodes may be structurally adjacent. Replacing scalar positions with Laplacian eigenvector coordinates $\phi\_v \in \mathbb{R}^k$ allows the attention mechanism to act as a kernel estimator of structural proximity (e.g., effective resistance or diffusion distance) [1], rather than sequential position.
>
> To isolate the contribution of spectral encoding richness, we ablate LapPE dimensionality from 16 to 8 (Weaver-8). Results show Weaver-8 degrades sharply on heterophilic benchmarks while homophilic datasets remain relatively stable. This heterophily-specific degradation, which would not arise if gains were purely text-driven, confirms that higher-frequency spectral features are critical for handling heterophily.
>
> Summary of extended results (for Table 2):
> |Model|Cora|Citeseer|History|Children|Sportsfit|WikiCS|Cornell|Texas|Wisconsin|Washington|
> |-|-|-|-|-|-|-|-|-|-|-|
> |Weaver|76.50|70.14|60.60|34.30|63.60|69.80|69.40|60.90|62.50|52.50|
> |Weaver-8|74.00|64.33|57.00|31.67|61.33|68.33|53.90|49.73|45.04|41.20|
>
> We acknowledge that a direct ablation comparing encoding strategies (RWPE, RRWP, or additive encoding) remains an important open comparison, which we list as a prioritized direction for future work.
>
> >Q2/W2: What exactly is the information bottleneck caused by external encoders?
>
> We thank the reviewer for this important question. The bottleneck in encoder-enhanced Graph LLMs operates at two levels: (i) textual compression, where node content spanning hundreds of tokens is reduced to a fixed number of memory tokens (e.g., 128 in GOFA, 1 in LLaGA), permanently discarding fine-grained textual details; and (ii) structural aggregation, where compressed representations are further aggregated over multi-hop neighborhoods, compounding information loss. This is formally proved by the over-squashing result [2], which shows that a $K$-hop GNN must compress information from an $O(\exp(K))$-sized receptive field into a fixed vector $h\_v \in \mathbb{R}^d$, mathematically exceeding its capacity.
>
> >Q3/W4: What is the key technical factor that makes Weaver effective compared to GOFA? Would adding an additional encoder to Weaver lead to a degradation in performance?
>
> The key technical factor is precisely the preservation of token-level message passing that aligns well with LLM pretrained manifold; additionally, the design of global attention helps avoid the over-squashing issue for neighbor-aggregating [2].
>
> To empirically verify our claim, we introduce a context encoder [3] that compresses each node's token sequence to 25% or 50% of its original length before passing it to Weaver, directly simulating the bottleneck introduced by external encoders.
>
> |Model|Cora|Citeseer|History|Children|Sportsfit|WikiCS|Cornell|Texas|Wisconsin|Washington|
> |-|-|-|-|-|-|-|-|-|-|-|
> |GOFA|71.06|65.72|56.25|12.15|37.87|68.62|39.50|38.37|32.51|31.02|
> |Weaver(0.25)|74.00|52.33|50.33|19.67|34.67|69.33|51.31|57.30|53.05|45.83|
> |Weaver(0.50)|76.33|54.33|54.85|26.67|38.33|72.67|52.36|50.27|56.87|47.69|
> |Weaver(full)|76.50|70.14|60.60|34.30|63.60|69.80|69.40|60.90|62.50|52.50|
>
> Despite compression-induced degradation, compressed Weaver variants tend to perform comparably to, and in many cases slightly better than, GOFA across most datasets. This marginal resilience may partially reflect the differences in graph encoding paradigm between the two models [3].
>
> We note that our current compression experiment uses a training-free encoder to simulate the bottleneck in a controlled manner. A more faithful comparison would involve training a dedicated ICAE-style encoder [4] jointly with Weaver, as GOFA does. However, joint encoder-LLM training of this kind requires on the order of 1,000+ GPU hours, which is prohibitive within the rebuttal period.
>
> >W3: Figure 1 caption and standard deviations
>
> We will expand the Figure 1 caption to describe each component. Regarding standard deviations, we will include standard deviation across independent runs with different random seeds in the revision.
>
> >W5: Lack of architecture comparison in main text
>
> We will add a concise comparison paragraph to Section 3.
>
> [1] Kreuzer et al. Rethinking Graph Transformers with Spectral Attention. NeurIPS 2021.
>
> [2] Alon et al. On the Bottleneck of Graph Neural Networks and its Practical Implications. ICLR 2021.
>
> [3] Pan et al. LLMLingua-2: Data Distillation for Efficient and Faithful Task-Agnostic Prompt Compression. Findings of ACL 2024.
>
> [4] Ge et al. In-context Autoencoder for Context Compression in a Large Language Model. ICLR 2024.

---

> > ### Author Rebuttal · Reviewer_g7N7 · 2026-04-01
> >
> > Thanks for your detailed response. My concerns are partially solved. I still have some additional questions as follows.
> >
> > As for **Q1&W1**, I commend that the authors use the additional experiments to address my concerns. Could you provide more theoretical evidence that RoPE makes the attention mechanism act as a kernel estimator of structural proximity rather than sequential position? This would better enhance the theoretical depth of the paper.
> >
> > As for **Q2&W2&Q3&W4**, thank you for addressing my concerns through the additional experiments. Could you provide more analysis regarding the efficiency comparison between GOFA and Weaver? As you mentioned, Weaver utilizes more node token sequences without compression. Is there a trade-off here between computational efficiency and model quality?
> >
> > As for **W3&W5**, thanks for addressing these.
> >
> > Overall, the additional experimental comparisons presented in the rebuttal have successfully resolved the majority of my concerns. I am keeping my positive rating and upgrading the Originality score from fair to good.

---

> > > ### Author Response · Authors · 2026-04-06
> > >
> > > We sincerely appreciate the reviewer's constructive feedback and the opportunity to provide additional clarification. We address each question below.
> > >
> > > > Could you provide more theoretical evidence that RoPE makes the attention mechanism act as a kernel of structural proximity?
> > >
> > > To clarify, let $\mathbf{q}, \mathbf{k} \in \mathbb{R}^d$ be query and key vectors whose components are i.i.d. with mean $\mu$ and variance $\sigma^2$ within each 2D subspace. For nodes $u, v$ with structural phase encodings $\boldsymbol{\phi}\_u, \boldsymbol{\phi}\_v \in \mathbb{R}^{d/2}$ repeated from truncated $p$-dim graph Laplacian eigenvectors, the expected attention logit satisfies:
> > > $$\mathbb{E}\_{\mathbf{q}, \mathbf{k}}\left[\mathbf{q}^\top R(u)^\top R(v) \mathbf{k}\right] = \alpha \sum\_{k=1}^{d/2} \cos\big(\frac{\hat{\theta}\_k}{f(\lambda\_k)}(\phi\_{u,k} - \phi\_{v,k})\big), $$
> > > where $\alpha = 2\mu^2$ when $\mathbf{q}$ and $\mathbf{k}$ are independent, or $\alpha = 2(\mu^2 + \sigma^2)$ when correlated (i.e., $\mathbf{k} = \mathbf{q} + \boldsymbol{\varepsilon}$ with $\mathbb{E}[\boldsymbol{\varepsilon}] = 0$), $f(\lambda\_k)$ is an eigenvalue encoding function.
> > >
> > > This indicates that the expected logit coincides with a shift-invariant structural kernel in the spectral embedding space. Under a neighborhood assumption, where structurally adjacent or similar nodes $u$ and $v$ satisfy $|\frac{\hat{\theta}\_k}{f(\lambda\_k)}(\phi\_{u,k} - \phi\_{v,k})| \ll 1$ for all $k$, the summation can be approximated by a second-order Taylor expansion:
> > > $$
> > > \mathbb{E}\_{\mathbf{q}, \mathbf{k}}\left[\mathbf{q}^\top R(u)^\top R(v) \mathbf{k}\right]
> > > = \alpha \left(\frac{d}{2} -  \hat{d}^2\_{\mathrm{struct}}(u,v)\right) + \mathcal{O}\left( \sum\_{k=1}^{d/2} \left(\frac{\hat{\theta}\_k}{f(\lambda\_k)} (\phi\_{u,k} - \phi\_{v,k})\right)^4 \right),
> > > $$
> > > where the induced structural distance is
> > > $\hat{d}^2\_{\mathrm{struct}}(u,v) = \frac{1}{2} \sum\_{k=1}^{d/2} \left(\frac{\hat{\theta}\_k}{f(\lambda\_k)}\right)^2 (\phi\_{u,k} - \phi\_{v,k})^2.$
> > > This term is a generalized squared spectral distance weighted by $\frac{\hat{\theta}\_k}{f(\lambda\_k)}$. Specifically, by fixing $\hat{\theta}\_k\equiv 1$ and varying $f(\lambda\_k)$, we can recover classical truncated spectral distances:
> > > diffusion distance [5] ($f(\lambda\_k) = e^{t\lambda\_k}$), biharmonic distance [6] ($f(\lambda\_k) = \lambda\_k$), and effective resistance [7] ($f(\lambda\_k) = \sqrt{\lambda\_k}$).
> > >
> > > Remark. For structurally distant nodes, large phase gaps generate high-frequency oscillations in the cosine terms. Under frequency diversity or incoherence assumptions, these oscillatory contributions tend to average out across modes, yielding a localized similarity on average; however, periodic aliasing remains possible.
> > >
> > > > Could you provide more analysis regarding the efficiency comparison between GOFA and Weaver? As you mentioned, Weaver utilizes more node token sequences without compression. Is there a trade-off here between computational efficiency and model quality?
> > >
> > > We fully agree that there is a fundamental trade-off between scalability and model quality, which we explicitly acknowledge in our Limitations section.
> > >
> > > For a concrete comparison, let $V$ denote the number of nodes, $E$ the number of edges, and $\bar{L}$ the average token length per graph element.
> > >
> > > - GOFA achieves linear scalability through compression and message-passing. Its dominant complexity arises from three stages: 1) ICAE compression of each text chunk into $K$ memory tokens: $\mathcal{O}\big((V + E)(\bar{L} + K)^2\big)$; 2) token-level message-passing: $\mathcal{O}\big(K(V + E)\big)$; 3) LLM prefilling with the target node's $K$ tokens: $\mathcal{O}(K^2)$. The overall complexity is dominated by $\mathcal{O}\big((V + E)(\bar{L} + K)^2\big)$.
> > >
> > > - Weaver forgoes compression to preserve full token-level semantics, applying global attention over the flattened graph sequence at a cost of $\mathcal{O}\big(((V + E)\bar{L})^2\big)$. Notably, for attribute-less graphs where edges carry no text, empty edges are simply omitted from the sequence, reducing complexity to $\mathcal{O}\big((V\bar{L})^2\big)$, whereas GOFA still process $K$ padding tokens per edge to maintain structural routing.
> > >
> > > In essence, Weaver currently trades scalability for uncompressed global reasoning, resulting in a complexity comparable to linearization-based LLMs. While our attention natively supports sparse masks that could alleviate this quadratic cost, rigorously evaluating the sparsity-quality trade-off is an important direction for future work.
> > >
> > > We hope these clarifications strengthen the practical understanding of our approach. Thank you again for your thorough review and valuable guidance in improving this work.
> > >
> > > [5] Coifman and Lafon. Diffusion Maps. Applied and Computational Harmonic Analysis 2006.
> > >
> > > [6] Lipman et al. Biharmonic Distance. ACM Trans. Graph. 2010.
> > >
> > > [7] Ellens et al. Effective Graph Resistance. Linear Algebra and Its Applications 2011.

---

### Official Review · Reviewer_ZPdJ · 2026-03-11

**Soundness:** 3
**Presentation:** 1
**Significance:** 3
**Originality:** 2
**Overall Recommendation:** 4
**Confidence:** 2

**Summary:**

The paper introduces Weaver, a framework that integrates graph topology with large language models (LLMs) for learning over text-attributed graphs. The method encodes the textual content associated with graph nodes and, optionally, edges as parallel token blocks.

To incorporate structural information, the model introduces structure-specific attention mechanisms that allow token representations to aggregate both intra-node information (within a node’s textual content) and inter-node information, with the latter being explicitly conditioned on the graph topology.

The framework proposes three main components:
(i) A graph masking strategy that enables topology-aware information propagation through the attention mechanism.
(ii) A graph-specific positional encoding designed to incorporate structural information.
(iii) A local prioritization mechanism aimed at breaking graph symmetries, particularly among nodes belonging to the same orbit partitions.

The method is evaluated primarily on node classification, link prediction, and a commonsense reasoning task formulated as a binary classification problem.

**Compliance With Llm Reviewing Policy:**

Affirmed.

**Final Justification:**

My main concerns were primarily related to the clarity of the presentation. While the authors indicate that they intend to improve the manuscript in this respect, it is difficult to assess the extent of these improvements without access to a revised version.

The rebuttal responses are relatively short, but it stems from the rebuttal format.
Overall, the authors have addressed all of my points, but it remains difficult to determine to what extent these clarifications would ultimately affect my evaluation.

Now that the author clarified their claim, I also raise some doubt about the significance of the task.

Please, note that I am not an expert in Graph2Text models, which explain my confidence score.

**Key Questions For Authors:**

What is the main task your approach is aiming to address?

**Limitations:**

The computational efficiency is an addressed limitation. However, we could regret that is it not better evaluated via an ablation study.

**Strengths And Weaknesses:**

## Strengths

- The paper builds on the observation that the main architectural differences between Transformers used in LLMs and Graph Transformers largely lie in the attention structure and positional encodings. While this observation is well known, the paper leverages it in a meaningful and principled way by reparameterizing attention visibility and positional structure to align with the input graph topology.
- The three proposed components, Graph Causal Masking, Graph-over-Tokens RoPE, and Local Prioritization, form a coherent and complementary set of design choices for adapting transformer architectures to graph-structured data. Together, they provide a structured way to incorporate graph topology into token-based models.
- The empirical results appear strong across the evaluated tasks. However, I am not sufficiently familiar with this specific experimental setting to fully assess whether the selected baselines are representative of the current state of the art.

---

## Weaknesses

- I found the paper is difficult to follow, primarily because the task and positioning of Weaver are not clearly defined. At first, the model appears to be presented as a graph-aware conversational system, as suggested by Figure 2, which depicts Weaver as a chat assistant operating over graphs, and by the framing of the method as an LLM-based approach. However, the experimental evaluation focuses almost exclusively on node classification and link prediction, which are standard graph learning tasks. The only text generation experiment (on the SceneGraphs dataset) is relegated to the appendix.
- Relatedly, the evaluation protocol for the text generation experiment is not clearly described. The paper reports an accuracy metric for answers to open-ended questions, but it is unclear how this metric is computed. Providing additional details on the evaluation procedure would improve clarity.
- Overall, the method appears closer to a text-attributed graph foundation model rather than a generative LLM system. Clarifying whether this characterization aligns with the authors’ intent would help better position the contribution. More generally, the status and intended scope of Weaver, whether it should be understood as a graph model, an LLM extension, or a hybrid foundation model, should be made explicit.
- Due to this ambiguity, the distinction between what the authors refer to as an LLM and what is essentially a graph transformer model is not always clear. As the paper correctly points out, the main architectural differences between text transformers and graph transformers lie in the attention structure and positional encodings, which makes the boundary between the two somewhat blurry. Providing a clearer definition of what is meant by “LLM” in this context, or using more precise terminology, would help avoid confusion.
- A central motivation for Weaver is to address the information bottleneck introduced by encoder-enhanced Graph LLM approaches. However, this bottleneck is only briefly mentioned and not sufficiently explained in the paper. Understanding this issue requires consulting the cited references. Providing a clearer explanation, such as describing the dimensionality of intermediate representations or how information compression occurs, would make the argument much easier to follow.

---

> ### Author Rebuttal · Authors · 2026-03-31
>
> Thank you very much for the constructive feedback and valuable suggestions for
> improving the paper's readability. We respond to each below.
>
> > W1,W3,W4/Q1: The task and positioning of Weaver are not clearly defined. What is the main task your approach is aiming to address?
>
> Thank you for this thorough analysis. The suggested clarification of task and positioning will indeed improve readability.
> We address the two core concerns below.
>
> Weaver aims to extend autoregressive LLMs to perform graph tasks as effectively as specialized graph foundation models, while retaining the LLM's language capabilities and cross-task transferability. We would like to clarify that, apart from SceneGraph QA, structural reasoning tasks (Figure 4) are also text generation tasks that remain challenging for current LLMs [1]. Their placement in the appendix unfortunately obscured this. Following the suggestions, the revision will:
> - Add SceneGraph results into Section 5 and add generation case studies in the Appendix. For reference, Weaver achieves 79.2% accuracy on SceneGraph QA, compared to 34.06% (GOFA, supervised performance from their report) and 66.0% (Qwen3-8B linearization). We attribute this in part to token-level access to node content rather than compressed graph-level representations.
> - Explicitly define "LLM" in our context (Section 1): an autoregressive model pretrained on large-scale corpora that possesses instruction-following ability, unifying diverse tasks under autoregressive generation. This helps distinguish Weaver from graph transformers, which lack pretrained language priors and require task-specific architectures rather than unifying diverse tasks within a single model.
>
> SceneGraph Results:
> | Model | Accuracy (%) |Δ vs. Weaver|
> |-|:-:|:-:|
> |GOFA|34.06|−45.14|
> |Qwen3-8B (Linearization)|66.0|−13.2|
> |Weaver|79.2|—|
>
> > W2: The evaluation protocol for the text generation experiment is not clearly described.
>
> Thank you for pointing this out. For SceneGraph QA, each question has a
> ground-truth label. We use GPT-5.4 as a judge to evaluate whether the generated
> answer semantically matches the ground truth, following standard LLM-as-judge
> protocols. We will add this description to the experiment setup section to better demonstrate Weaver's generative capabilities.
>
> > W5: The information bottleneck is not sufficiently explained.
>
> We agree this deserves a clearer treatment. The bottleneck arises in encoder-enhanced Graph LLMs (e.g., GOFA, LLaGA), where each node's textual content is first compressed into a small number of memory tokens by a language model encoder, and these compressed representations are then aggregated by a graph encoder before being passed to the LLM. For example, a node whose text spans hundreds of tokens may be reduced to only a few fixed-length memory tokens (e.g., 128 in GOFA and 1 in LLaGA), inevitably losing fine-grained textual details. Weaver avoids this compression by representing each node as its full token sequence within the LLM attention, allowing the model to attend to individual tokens across nodes. We will add this explanation, together with a concrete token-length example, to Section 1.
>
> > Limitation: The computational efficiency not better evaluated via ablation.
>
> Thank you for the suggestion. We provide an efficiency ablation over node text length. Weaver shows relatively stable TTFT as text length increases (20 nodes, from 100 to 800 chars per node), which we attribute to its prefilling-based design: graph context is incorporated directly into the LLM and is largely parallelizable during prefilling. By contrast, GOFA relies on encoder-side neighbor aggregation, which is less parallelizable and therefore incurs substantially higher latency. LLaGA is the most efficient because it compresses each node, regardless of text length, into a single token before the LLM. This reduces latency and memory, but also creates a stronger information bottleneck, which may contribute to its lower performance in Table 2.
>
> |Model|TTFT (ms)@100|TTFT (ms)@200|TTFT (ms)@400|TTFT (ms)@800|Mem (GB)@100|Mem (GB)@200|Mem (GB)@400|Mem (GB)@800|
> |-|-|-|-|-|-|-|-|-|
> |LLaGA (w/o encode)|65.6±5.4|50.0±2.8|59.8±9.5|64.7±6.9|13.0±0.0|13.0±0.0|13.1±0.0|13.1±0.0|
> |GOFA|1015.8±1.5|1066.5±1.4|1174.8±1.3|1439.3±2.2|22.4±0.0|22.7±0.0|23.4±0.0|24.5±0.0|
> |Qwen3(flash-attention)|134.5±13.9|219.0±14.1|322.4±11.0|635.3±10.7|15.7±0.0|16.0±0.0|16.7±0.0|18.0±0.0|
> |Qwen3(sdpa)|134.6±14.7|203.0±17.1|325.6±9.8|635.2±16.0|15.7±0.0|16.0±0.0|16.7±0.0|18.0±0.0|
> |Weaver(eager)|144.8±11.9|282.9±5.2|666.3±24.7|1862.1±33.8|15.6±0.0|16.3±0.0|18.9±0.0|29.0±0.0|
> |Weaver(flex-attention)|133.3±8.7|192.2±7.1|378.5±9.9|796.6±33.8|15.6±0.0|15.9±0.0|16.5±0.0|17.8±0.0|
>
>
> [1] Xu et al. GraphOmni: A Comprehensive and Extensible Benchmark Framework for Large Language Models on Graph-theoretic Tasks. ICLR 2026.

---

> > ### Author Rebuttal · Reviewer_ZPdJ · 2026-04-03
> >
> > Thank you to the authors for their response. My main concerns were primarily related to the clarity of the presentation. While the authors indicate that they intend to improve the manuscript in this respect, it is difficult to assess the extent of these improvements without access to a revised version.
> >
> > The rebuttal responses are relatively brief, and in several cases addressing my concerns would likely require more substantial changes to the paper itself. At the same time, I understand that the rebuttal format imposes strict character limitations.
> >
> > Overall, I view this as a limitation of the review process rather than of the authors’ response itself. The authors have addressed all of my points, but it remains difficult to determine to what extent these clarifications would ultimately affect my evaluation.

---

### Official Review · Reviewer_kAHz · 2026-03-12

**Soundness:** 3
**Presentation:** 2
**Significance:** 3
**Originality:** 3
**Overall Recommendation:** 4
**Confidence:** 3

**Summary:**

n this paper, the authors present Weaver, which designs a generative graph language model by modifying the causal attention in LLMs to introduce graph bias. Specifically, in Weaver: 1. A graph-causal masking mechanism is added to ensure causal masking within each node but bidirectional masking between nodes for a global view. 2. A graph-over-token RoPE is introduced using LapPE to incorporate structural bias. 3. Local prioritization is designed to ensure the model favors localization.

**Compliance With Llm Reviewing Policy:**

Affirmed.

**Final Justification:**

The author's rebuttal addresses most of my concerns, and I personally think the proposed method is a novel direction for GFM development and vote for acceptance.

**Key Questions For Authors:**

1. I did not fully understand how Weaver incorporates edge attributes. What are the attention mask and RoPE settings for the edge tokens?
2. Regarding the local prioritization, it seems to simply penalize nodes that are far from each other after sequentialization. What if two nodes are connected but become far apart after sequentialization? Furthermore, since all edges are appended after the nodes, a particular edge token can be far away from its source and target nodes, which may be improperly penalized by this term. Please clarify it if I misunderstood it.

**Limitations:**

See above.

**Strengths And Weaknesses:**

## Strengths

1. Overall, I think the proposed method is interesting and makes sense.

2. The authors provide extensive experiments to validate the effectiveness of their method.

## Weaknesses
1. The computation of GoRoPE requires LapPE, which has a complexity of cube n. This may limit its application on large graphs.
2. Since the design of Weaver favors node-level tasks, how does Weaver perform on edge-attribute-heavy tasks, such as knowledge graph completion?
3. The base model for Weaver is Qwen3, which differs from many baseline models (such as GOFA or LLaGA). It would be better to normalize the base model choice to isolate the effect of Weaver’s masking and positional encoding design.
4. Can the authors provide an ablation study or computational cost analysis (in terms of memory and time) for the LapPE component?
5. Can the authors provide runtime comparison (both training and inference) between Weaver and other baseline GLMs.

---

> ### Author Rebuttal · Authors · 2026-03-31
>
> Thank you for the constructive and encouraging feedback. We address your questions and concerns below.
>
> >Q1: How does Weaver incorporate edge attributes?
>
> Edges are modeled as independent token blocks (like nodes) with node-derived GoRoPE. Intra-edge tokens use causal masking; all cross-block attention (edge-node, edge-edge) is bidirectional. As defined in Eq. 10, each token in a directed edge block is assigned a structural coordinate formed by concatenating the LapPE encodings of its two endpoints, explicitly encoding both endpoints and directionality.
>
> >Q2: Regarding the local prioritization ...
>
> Thank you for pointing this out! To clarify: although local prioritization reuses the ALiBi slope scheduler, the underlying mask is **binary**, not distance-graded. As shown in Eq. 12 (and visualized in Figure 1), each position is labeled either 1 (intra-block) or 0 (inter-block), and the attention bias is then the slope times this binary indicator. Within each head, all cross-block attention is therefore equally attenuated, serving only to prioritize the establishment of intra-block semantics before composing cross-block context. Crucially, this makes the module still independent of serialization order, as no token is penalized more for appearing later in the prompt.
>
> We acknowledge that calling it an "ALiBi slope scheduler" without this clarification can misleadingly suggest a diagonal-centered distance decay. We will add an note in the manuscript to prevent this confusion.
>
> >W1: The computation of GoRoPE requires LapPE ... may limit its application on large graphs.
>
> We thank the reviewer for this practical point. We clarify that LapPE only requires the bottom-$k$ eigenvectors (e.g., $k{=}16$ in our experiments), which can be computed via a sparse solver such as ARPACK (Lanczos) in $O(k \cdot \mathrm{nnz})$ time. For large graphs where even this becomes costly, randomised proxies such as FastRP can bypass eigendecomposition entirely.
>
> >W2: How does Weaver perform on edge-attribute-heavy tasks?
>
> This is a challenging scenario for current GLMs. A 41-node SceneGraph with 31% edge density causes GOFA to OOM on A100-80GB; Weaver requires only 16.84GB with FlexAttention. To scale further, Weaver can use sparse masks where edge blocks serve as prefilled KV-caches queried only by node tokens. Letting $n$ = number of nodes, $m$ = number of edges, $L\_n$ = tokens per node block, and $L\_e$ = tokens per edge block, this reduces complexity from $\mathcal{O}(n^2L\_n^2 + nm L\_nL\_e + m^2L\_e^2)$ to $\mathcal{O}(n^2L\_n^2 + nm L\_nL\_e + m L\_e^2)$, eliminating the dominant $m^2L\_e^2$ edge-edge cross-attention term while preserving full node-node and node-edge interactions.
>
> >W3: It would be better to normalize the base model to isolate the effect of Weaver’s design.
>
> We agree that aligning the base model would strengthen our claims. However, we found this procedure to be prohibitively expensive. For example, retraining the ICAE (In-Context Auto-Encoder) that GOFA relies on alone requires over 1,000 GPU hours, making full re-implementation on Qwen3 infeasible within the rebuttal period.
>
> Alternatively, we measure each method's gain over its own base model on SceneGraph:
>
> | Model | Accuracy (%) |Δ vs Base|
> |-|:-:|:-:|
> |Mistral|45.95|−|
> |GOFA|34.06|-11.89|
> |Qwen3-8B|66.0|-|
> |Weaver|79.2|+13.2|
>
> While this delta metric cannot fully disentangle architecture from base-model strength, the key observation is directional: GOFA actively degrades Mistral (−11.89) despite its lower baseline leaving more room to improve, whereas Weaver yields a substantial +13.2% gain on top of an already strong Qwen3 baseline where less headroom remains, suggesting the improvement stems from architectural design rather than base-model strength alone.
>
> >W4: Can the authors provide an ablation study or computational cost analysis for the LapPE component?
>
> The $O(n^3)$ figure is the dense full-spectrum worst case. Practically, LapPE only needs bottom-$k$ eigenvectors, reducing cost to $O(k \cdot \mathrm{nnz})$ via Lanczos, plus $O(kn\log n)$ for canonization.
>
> We provide empirical latency (ms) on Erdős–Rényi random graphs with 30% edge density:
> |Method|N=100|N=400|N=1000|
> |-|:-:|:-:|:-:|
> |RWPE|0.6|8.5|1.9|
> |LapPE|4.3|29.0|208.0|
> |Canonized LapPE|77.1|295.7|697.7|
>
> Even at N=1000, canonized LapPE takes under 0.7s, a one-time preprocessing cost that is amortized across training and negligible relative to LLM forward passes (typically seconds per graph). We will include this analysis in the revision.
>
> >W5: Can the authors provide runtime comparison between Weaver and other baselines?
>
> We are pleased to provide the runtime comparison and kindly refer the reviewer to our response to the last weakness of Reviewer ZPdJ, where a detailed comparison is provided due to character limitations here.

---

> > ### Author Rebuttal · Reviewer_kAHz · 2026-04-03
> >
> > I thank the author for the detailed response to all my concerns. Most of my concerns have been resolved, and I will keep my score.

---

### Decision · Program_Chairs · 2026-04-30

**Decision:**

Accept (regular)

**Comment:**

This paper introduces Weaver, an encoder-free framework for incorporating graph structure into decoder-only LLMs. The key idea is to reinterpret attention as message passing and to embed the graph topology directly into the attention mechanism.

Reviewers acknowledged that, by directly modifying attention and positional encoding, the method provides a principled way to integrate graph topology into autoregressive LLMs without requiring external graph encoders. They also acknowledged that the proposed work demonstrates strong empirical performance.

On the other hand, the concerns include:
- GoRoPE relies on Laplacian Eigenmaps (LapPE), which technically has a cubic complexity. While authors clarified that Lanczos solvers make this manageable, it remains a non-trivial step for extremely large graphs.
- Reviewers initially found the positioning of Weaver (between a generative assistant and a discriminative graph model) ambiguous. The evaluation metrics for the generative task (SceneGraph) were also initially underdescribed.
- Comparative experiments used different base LLMs (Mistral vs. Qwen3), making it difficult to perfectly isolate the architecture's contribution from the base model's strength.